# White matter myelination during early infancy is linked to spatial gradients and myelin content at birth

Mareike Grotheer [1,2✉], Mona Rosenke[3], Hua Wu [4], Holly Kular[3], Francesca R. Querdasi[3], Vaidehi S. Natu [3], Jason D. Yeatman[3,5,6,7] & Kalanit Grill-Spector [3,5]

Development of myelin, a fatty sheath that insulates nerve fibers, is critical for brain function. Myelination during infancy has been studied with histology, but postmortem data cannot evaluate the longitudinal trajectory of white matter development. Here, we obtained longitudinal diffusion MRI and quantitative MRI measures of longitudinal relaxation rate (R1) of white matter in 0, 3 and 6 months-old human infants, and developed an automated method to identify white matter bundles and quantify their properties in each infant's brain. We find that R1 increases from newborns to 6-months-olds in all bundles. R1 development is non-uniform: there is faster development in white matter that is less mature in newborns, and development rate increases along inferior-to-superior as well as anterior-to-posterior spatial gradients. As R1 is linearly related to myelin fraction in white matter bundles, these findings open new avenues to elucidate typical and atypical white matter myelination in early infancy.

[1] Department of Psychology, Philipps-Universität Marburg, Marburg 35039, Germany. [2] Center for Mind, Brain and Behavior – CMBB, Philipps-Universität Marburg and Justus-Liebig-Universität Giessen, Marburg 35039, Germany. [3] Department of Psychology, Stanford University, Stanford, CA 94305, USA. [4] Cognitive and Neurobiological Imaging Center (CNI), Stanford University, Stanford, CA 94305, USA. [5] Wu Tsai Neurosciences Institute, Stanford University, Stanford, CA 94305, USA. [6] Graduate School of Education, Stanford University, Stanford, CA 94305, USA. [7] Division of Developmental-Behavioral Pediatrics, Stanford University School of Medicine, Stanford, CA 94305, USA. ✉email: mareike.grotheer@uni-marburg.de

D uring the first year of life, the volume of the human brain's white matter increases by 6–16%[1]. A key microstructural component of this white matter development is myelination[2–6]. That is, the formation of myelin, the fatty sheath that insulates axons that connect different brain regions. Myelin is essential for brain function, as it enables rapid and synchronized neural communication across the brain and abnormalities in myelination are linked to a plethora of developmental and cognitive disorders[7]. However, the principles and nature of white matter myelination of the human brain during early infancy are not well understood.

Three main theories of white matter myelin development during infancy have been proposed: (1) The starts-first/finishes-first hypothesis, which is based on data from classic histological studies[2–4], proposes that postnatal myelination follows prenatal patterns. This hypothesis predicts that white matter that is more myelinated at birth will develop faster postnatally and will finish myelinating earlier. This, in turn, may allow for the most important brain functions to mature the fastest. (2) The speed-up hypothesis, which is based on more recent imaging data[8,9], suggests that white matter that is less myelinated at birth develops faster postnatally. This development may be experience-dependent[10–13] and may foster the efficient and coordinated transmission of signals across the entire brain. Both of the above hypotheses build on the observation that myelin content is not homogenous in the newborn brain[2–5,14]. (3) The spatial-gradient hypothesis suggests that postnatal myelination progresses in a spatially organized manner[5,15]. Different spatial gradients of myelination have been proposed, including that white matter myelination begins closest to the neurons and follows the direction of information flow[4], or that it occurs along a proximal to distal axis across the brain[5]. It is important to note that, while the starts-first/finishes-first hypothesis and the speed-up hypothesis are mutually exclusive, spatial gradients may contribute to myelination during infancy in addition to the effects of myelin content at birth predicted by the former two hypotheses.

Testing these developmental hypotheses requires in vivo measurements of the typical, longitudinal development of myelin along the length of multiple white matter bundles of individual infants. However, classic histological studies compare postmortem brain samples across individuals, often include pathologies, and use observer-dependent methods[16]. Thus, classic histology provides a cross-sectional and qualitative glimpse of white matter myelination during infancy. Up to recently[17–22] most in vivo investigations of white matter development leveraged diffusion metrics (e.g., mean diffusivity (MD)), that have a complex, non-linear relationship to myelin and are also affected by other properties of the white matter, including the diameter, spacing, and orientation of fibers[18,23–25]. Thus, diffusion metrics do not provide accurate measures of myelination. However, quantitative MRI[9,14,15,18,26–30] (qMRI) measurements, such as the longitudinal relaxation rate, R1 [s$^{-1}$], now offer metrics that are directly related to myelin content in the white matter. In fact, not only does the amount of myelin in a voxel (myelin fraction) explain 90% of the variance in R1 in white matter bundles[29,31], but also there is a linear relation between myelin fraction and R1 (Supplementary Fig. 1). Thus, longitudinal measurements of R1 along white matter bundles enable the assessment of white matter myelin development during infancy.

To test the predictions of the developmental hypotheses of white matter myelination during early infancy, we acquired longitudinal measurements of anatomical MRI, diffusion MRI (dMRI), and qMRI in infants during natural sleep at 3 timepoints: newborn ($N = 9$; age: 8–37 days), 3 months ($N = 10$; age: 79–106 days), and 6 months ($N = 10$; age: 167–195 days) of age.

We used anatomical MRI to segment the brain to gray and white matter, dMRI to identify the white matter bundles of the infant brain, and qMRI to measure R1 along each white matter bundle (Supplementary Fig. 2). All analyses were performed in each infant's native brain space. To relate our findings to prior developmental studies, we also used dMRI data to assess the development of MD in white matter bundles. However, as the relationship between MD and myelin is complex and nonlinear, we cannot accurately estimate from the rate of MD development the rate of myelination[9].

As increases in myelin in the white matter generate linear increases in R1, the developmental hypotheses tested here make the following predictions: The starts-first/finishes-first hypothesis predicts that during the first 6 months of life, R1 will increase faster in white matter that is more myelinated at birth and hence has higher R1 values in newborns. The speed-up hypothesis predicts the opposite, that during the first 6 months of life, R1 will increase faster in white matter that has lower R1 values in newborns. Finally, the spatial gradient hypothesis predicts spatially continuous differences in the development of R1 across the white matter, which cannot be explained by differences in R1 values in newborns.

Here we show that R1 of white matter bundles increases from newborns to 6-months-olds and that this development is non-uniform within and across bundles. That is, we find faster R1 development in sections of bundles that are less mature in newborns, consistent with the predictions of the speed-up hypothesis of infant myelination. In addition, we find that the rate of R1 development increases along the inferior-to-superior and anterior-to-posterior axes, consistent with the spatial gradient hypothesis. Thus, our findings suggest that myelination of human white matter bundles during early infancy is linked to both the initial myelin content at birth and spatial gradients.

## Results

### A new method for automated fiber quantification in babies (babyAFQ).
We first identified each individual infant's white matter bundles in their native brain space in a systematic and automated way. A major challenge is that present automated tools for bundle identification in individuals (e.g.[32–34]) have been developed for adults and school-aged children and therefore may not be suitable for infants due to substantial differences in brain size[1] and organization[20]. Thus, we developed a pipeline for analyzing infant dMRI data (Supplementary Fig. 2) and a novel method, baby automated fiber quantification (babyAFQ), for automatically identifying 24 bundles (11 in each hemisphere and 2 between-hemispheres, Supplementary Figs. 2–4) in each individual infant's brain and timepoint (Supplementary Fig. 8). We optimized babyAFQ for infants by: (i) generating waypoints (anatomical regions of interest (ROIs) for defining bundles) on a newborn brain template (University of North Carolina (UNC) neonatal template[35]), (ii) decreasing the spatial extent of waypoints compared to adults[36] to fit the more compact infant brain, (iii) adding waypoints for curved bundles to improve their identification, and (iv) offering a volumetric approach for the identification of the vertical occipital fasciculus (VOF) (Supplementary Fig. 4), as the VOF is often identified using cortical surface ROIs and cortical surface reconstructions can be difficult to generate for infant brains.

BabyAFQ successfully identifies 24 bundles in each infant and timepoint (example infant: Fig. 1, all infants: Supplementary Fig. 8), including bundles that have not previously been identified in infants: the posterior arcuate fasciculus[37], vertical occipital fasciculus[37–39], and middle longitudinal fasciculus[40]. The 24 bundles have the expected shape and location in all

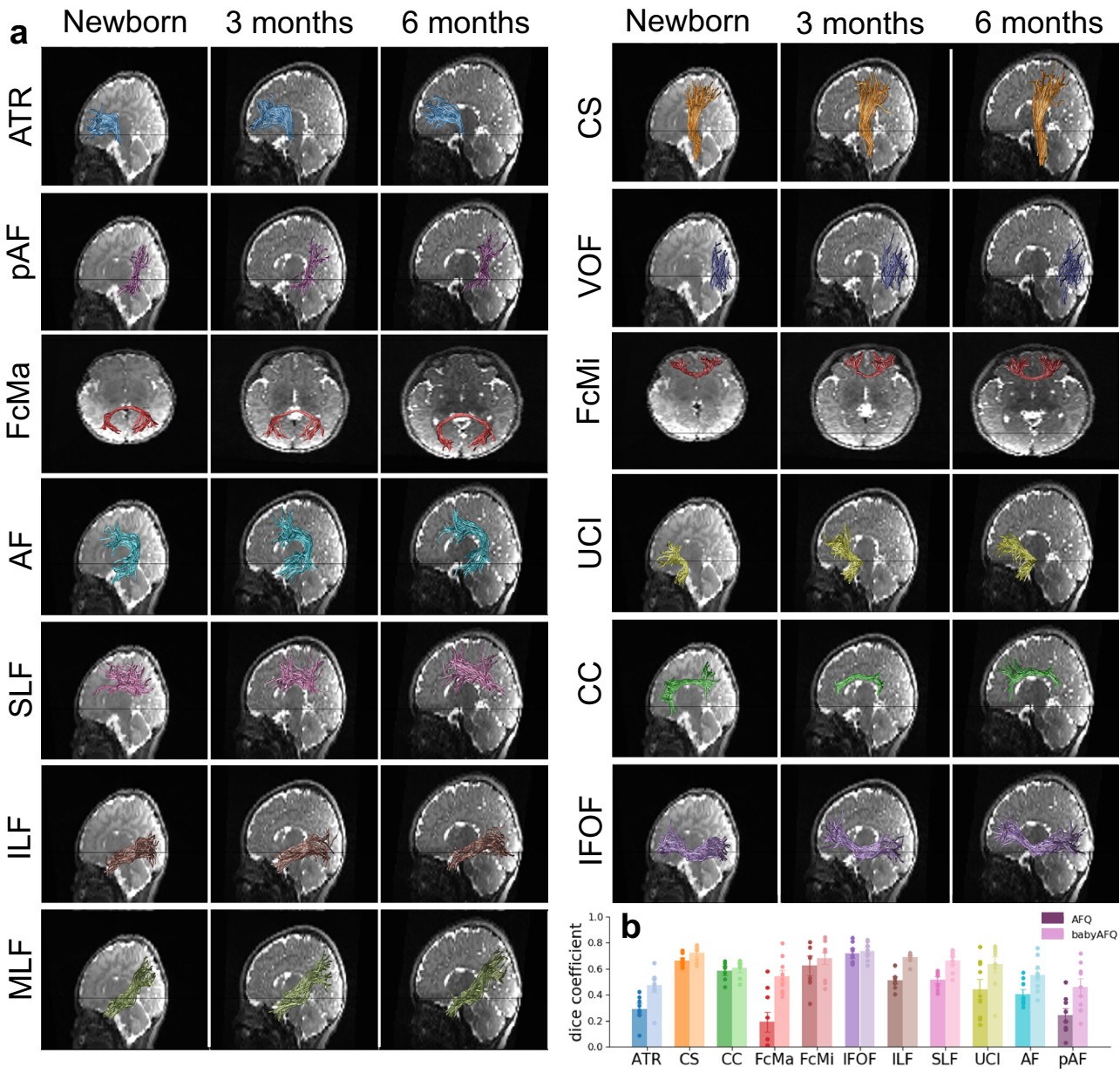

**Fig. 1 Baby automated fiber quantification (babyAFQ) identifies white matter bundles in individual infant brains across the first 6 months of life.**
Twenty-four bundles (11 in each hemisphere and 2 cross-hemispheric) were successfully identified in all individuals and ages (Supplementary Fig. 8). **a** All bundles of an example individual infant. Each row is a bundle, each column is a timepoint; left: newborn, middle: 3 months, right: 6 months. **b** Comparison of AFQ and babyAFQ performances in identifying each bundle in newborns ($N = 9$) relative to manually defined (gold-standard) bundles. Overlap between automatically and manually defined bundles is evaluated using the dice coefficient, which reveals higher performance for babyAFQ than AFQ. Bars show mean dice coefficient ± standard error across participants; circles: individual data. Source data are provided as a Source Data file. ATR anterior thalamic radiation, CS cortico-spinal tract, pAF posterior arcuate fasciculus, VOF vertical occipital fasciculus, FcMa forceps major, FcMi forceps minor, AF arcuate fasciculus, UCI uncinate fasciculus, SLF superior longitudinal fasciculus, CC cingulum cingulate, ILF inferior longitudinal fasciculus, IFOF inferior frontal occipital fasciculus, MLF middle longitudinal fasciculus.

infants even as their brains grow from 0 to 6 months. 3D interactive visualizations at 0 months (http://vpnl.stanford.edu/babyAFQ/bb11_mri0_interactive.html), 3 months (http://vpnl.stanford.edu/babyAFQ/bb11_mri3_interactive.html) and 6 months of age (http://vpnl.stanford.edu/babyAFQ/bb11_mri6_interactive.html) show the 3D structure of bundles in an example infant.

For quality assurance, we compared babyAFQ and AFQ[32] (developed in adults and used in prior infant studies[41–43]) to manually identified bundles ('gold-standard'). In newborns, bundles identified by babyAFQ substantially overlapped the gold-standard (mean dice coefficient ± standard error (SE):

$0.61 ± 0.02$) and this overlap was significantly higher compared to AFQ (Fig. 1b; Supplementary Fig. 3; two-way repeated measure analysis of variance (rmANOVA) with AFQ-type and bundle as factors: AFQ-type: $F(1,08) = 528.60$, $p < 0.0001$, bundle: $F(19,152) = 11.31$, $p < 0.0001$, AFQ-type × bundle: $F(19,152) = 7.13$, $p < 0.0001$; additional three-way rmANOVA on the bilateral bundles, with AFQ-type, bundle, and hemisphere as factors revealed no effects of, or interaction with, hemisphere). The VOF and MLF were not included in this comparison to manual bundles; this is because the MLF is not identified by AFQ and the VOF is identified using a different

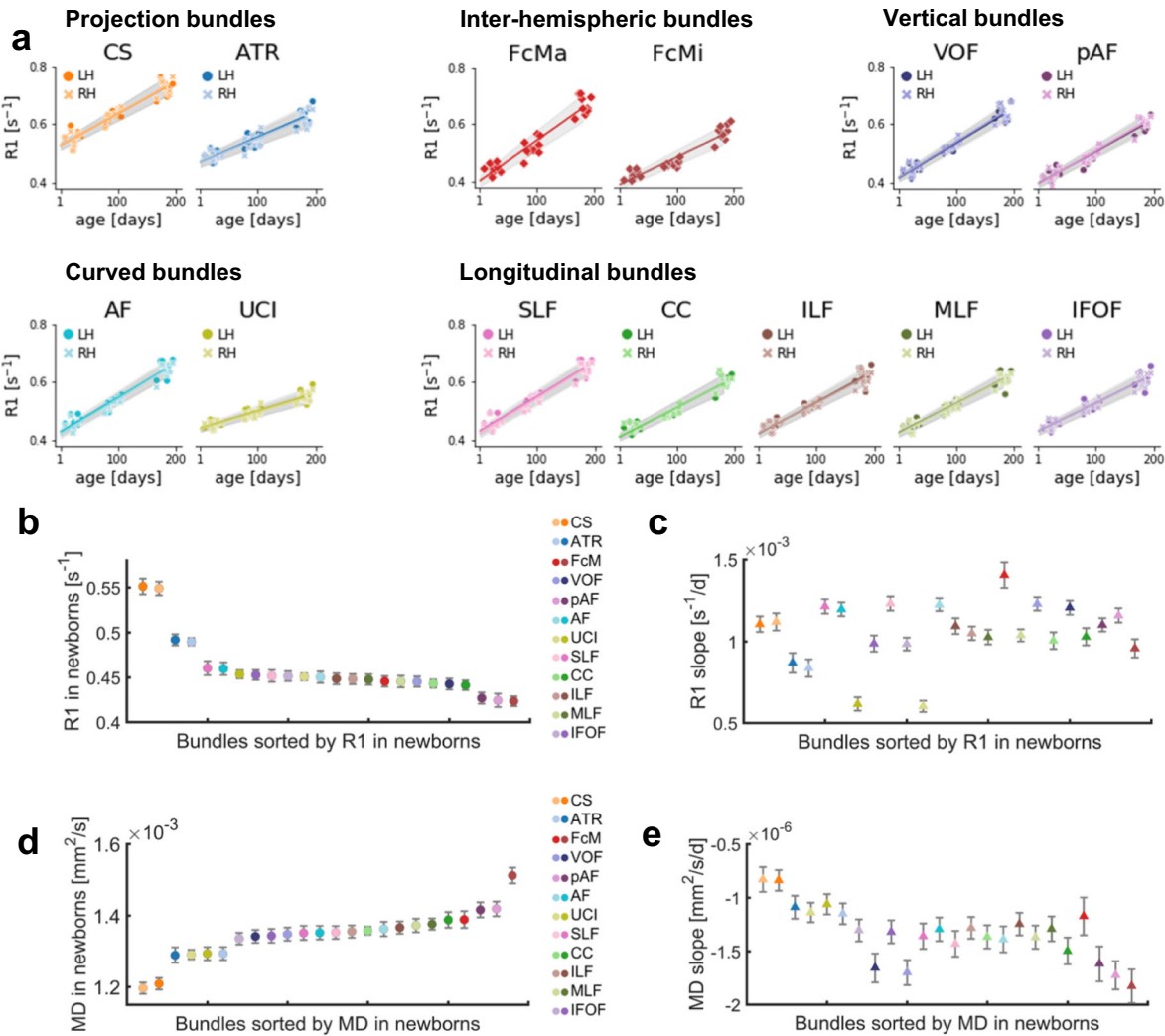

**Fig. 2 Mean R1 and mean MD of white matter bundles develop linearly from birth to 6 months of age. a** Mean R1 of each bundle as a function of age in days. Each point is a participant; markers indicate hemisphere; lines indicate LMM prediction; lines for both hemispheres fall on top of each other; gray shaded regions indicate 95% confidence intervals. **b** Mean R1 measured in newborns ($N = 9$) for 24 white matter bundles. **c** Rate of mean R1 development (slopes from LMMs) during the first 6 months of life for each white matter bundle; Bundles are sorted by R1 at birth. **d** Mean MD measured in newborns ($N = 9$) for 24 white matter bundles. **e** Mean MD decreases linearly from 0 to 6 months, which can be modeled by LMMs (Supplementary Fig. 5). Here we show the rate of mean MD development (LMM slopes) during the first 6 months of life for each white matter bundle. Note that slopes are negative; Bundles are sorted by MD at birth. **b**–**e** Color: bundle; Darker shades: LH. Error bars: Standard error. Source data are provided as a Source Data file. CS cortico-spinal tract, ATR anterior thalamic radiation, FcMa forceps major, FcMi forceps minor, VOF vertical occipital fasciculus, pAF posterior arcuate fasciculus, AF arcuate fasciculus, UCI uncinate fasciculus, SLF superior longitudinal fasciculus, CC cingulum cingulate, ILF inferior longitudinal fasciculus, MLF middle longitudinal fasciculus, IFOF inferior frontal occipital fasciculus, RH right hemisphere, LH left hemisphere.

methodological approach in AFQ (for details see Supplementary Fig. 4). Improvements from babyAFQ were also evident at the other timepoints in qualitative evaluations in individual infants. E.g., the Forceps Major was successfully identified by babyAFQ in 29/29 brains, but identified by AFQ in only 13/29 brains (Supplementary Fig. 8).

**During infancy, R1 increases in all 24 evaluated white matter bundles.** We first measured the development of mean R1 in each bundle during the first 6 months of life. Measurements of mean R1 of the 24 bundles identified by babyAFQ at 0, 3, and 6 months reveal a substantial increase in R1 from 0 to 6 months of age (Fig. 2a). Mean R1 across bundles ± SE [range]: 0 months: $0.46\,\mathrm{s}^{-1} \pm 0.007\,\mathrm{s}^{-1}$ [$0.42$–$0.55\,\mathrm{s}^{-1}$], 3 months: $0.52\,\mathrm{s}^{-1} \pm 0.008\,\mathrm{s}^{-1}$ [$0.46$–$0.63\,\mathrm{s}^{-1}$], 6 months: $0.62\,\mathrm{s}^{-1} \pm 0.009\,\mathrm{s}^{-1}$ [$0.54$–$0.73\,\mathrm{s}^{-1}$]. This is a profound change, as mean R1 increases on average by ~17% ($0.16\,\mathrm{s}^{-1}$)

within just 6 months. We modeled mean R1 development in each bundle using linear mixed models (LMMs) with age as predictor and a random intercept (estimated R1 at birth) for each participant. Overall, LMMs explained ~90% of the R1 variance across development (adjusted $Rs^2 > 0.87$, $ps < 0.0001$). As R1 in white matter is linearly related to myelin fraction, these data are consistent with the idea that white matter bundles myelinate during early infancy. To summarize the LMM results we plotted each bundle's mean R1 measured in newborns (Fig. 2b) and its rate of development (Fig. 2c) with three notable findings: (i) Mean R1 measured in newborns varies across bundles. At birth, projection bundles (CS and ATR) have the highest R1, and the forceps minor (FMi) and inferior frontal occipital fasciculus (IFOF) have the lowest R1 (Fig. 2b). (ii) The rate of R1 development during infancy varies between bundles. Across the 24 bundles, the forceps major (FcMa) has the fastest rate of R1 development, while the uncinate (UCI) and the anterior thalamic radiation (ATR) have the

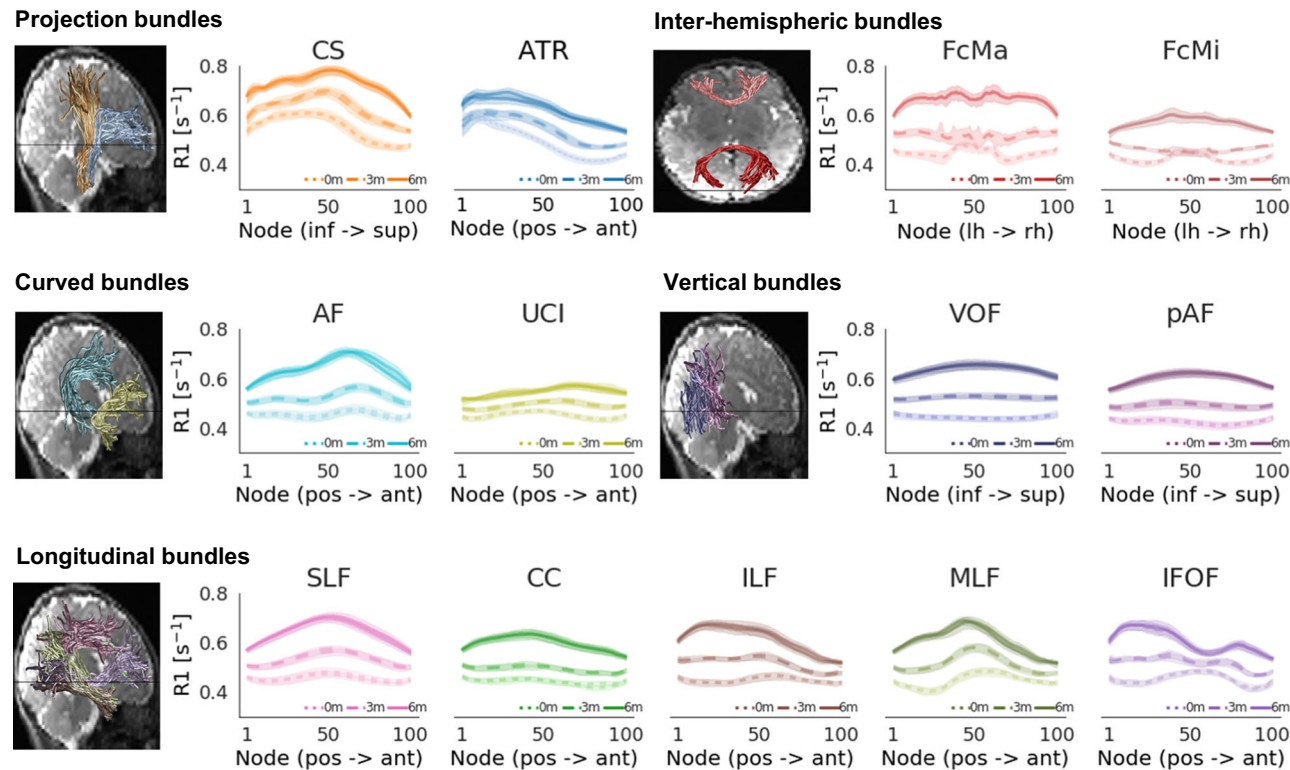

**Fig. 3 Development of R1 along each bundle.** R1 along the length of each bundle in newborns (0 m, dotted line), 3-months-olds (3 m, dashed line), and 6-months-olds (6 m, solid line). Lines: average R1 at each node across participants. Lines per hemisphere largely overlap. Shaded regions: 95% confidence intervals. Left panels show the bundles in a representative newborn. Source data are provided as a Source Data file. CS cortico-spinal tract, ATR anterior thalamic radiation, FcMa forceps major, FcMi forceps minor, VOF vertical occipital fasciculus, pAF posterior arcuate fasciculus, AF arcuate fasciculus, UCI uncinate fasciculus, SLF superior longitudinal fasciculus, CC cingulum cingulate, ILF inferior longitudinal fasciculus, MLF middle longitudinal fasciculus, IFOF inferior frontal occipital fasciculus.

slowest rate of R1 development between 0 to 6 months. (iii) Relating the bundles' rate of R1 development to their R1 measured in newborns reveals no systematic relationship between mean R1 in newborns and rate of mean R1 development (Fig. 2c). Indeed, there is no significant correlation between R1 in newborns and R1 slopes across bundles ($R^2 = 0.003$, $p = 0.81$). For example, both the cortical spinal tract (CS) and the forceps major (FcMa) have fast R1 development (steep slopes) during early infancy, yet they have vastly different mean R1 in newborns. Together, these analyses suggest that mean R1 in newborns does not seem to explain mean R1 development rate during early infancy.

To relate our findings to previous work that evaluated diffusion metrics[17–22], we also measured the development of MD across bundles. Myelination of the white matter is expected to result in decreases in MD. Consistent with this, we found that mean MD systematically decreases in all 24 white matter bundles during the first 6 months of life (Supplementary Fig. 5). Like R1, mean MD in newborns and the rate of mean MD development varied across bundles (Fig. 2d, e). Interestingly, while mean MD and R1 in newborns are correlated ($R^2 = 0.76$, $p < 0.0001$), the rates of MD and R1 development during early infancy are not correlated ($R^2 = 0.08$, $p = 0.17$) across bundles. That is, the longitudinal developmental patterns observed using MD are different from those observed with R1. For example, the uncinate (UCI) has slow R1 development (shallow slope) but rapid MD development (steep slope). In contrast to R1, we find a negative correlation between the rate of MD development and the measured MD in newborns ($R^2 = 0.71$, $p < 0.0001$), such that bundles with higher mean MD in newborns have an accelerated decrease in MD during early infancy. The differential development of MD and R1

is consistent with prior reports across the lifespan[44] and suggests that other changes to the white matter beyond myelination contribute to MD development in the first 6 months of life.

**R1 development during early infancy varies along the length of white matter bundles.** White matter bundles are large structures that span substantial distances across the brain and have variable white matter properties along their length[32,44]. Thus, mean measurements across the entire bundle may not be representative and may even obscure differential development patterns along the length of the bundles. Thus, we next evaluated R1 development along the length of 24 bundles.

We examined the development of R1 along each bundle using babyAFQ with two main observations: (i) At each timepoint, R1 exhibits spatial variations along the length of the 24 bundles (Fig. 3), with the range of variations differing across bundles. For example, the cortico-spinal tract (CS) exhibits substantial variations in R1 along its length, whereas the vertical occipital fasciculus (VOF) shows only modest variations. (ii) Consistent with the analyses of mean R1, along the length of each of the 24 bundles, R1 systematically increases from newborns (Fig. 3-dotted line), to 3-month-olds (Fig. 3-dashed line), to 6-months-olds (Fig. 3-solid line).

To quantify R1 development along white matter bundles during the first 6 months of life, we used LMMs applied independently at 100 equidistant locations (nodes) along each bundle (LMM relating R1 to age; one LMM per node and bundle; random intercepts for individuals). The LMM slopes estimate the rate of R1 development at each node (Fig. 4-dashed lines), and we

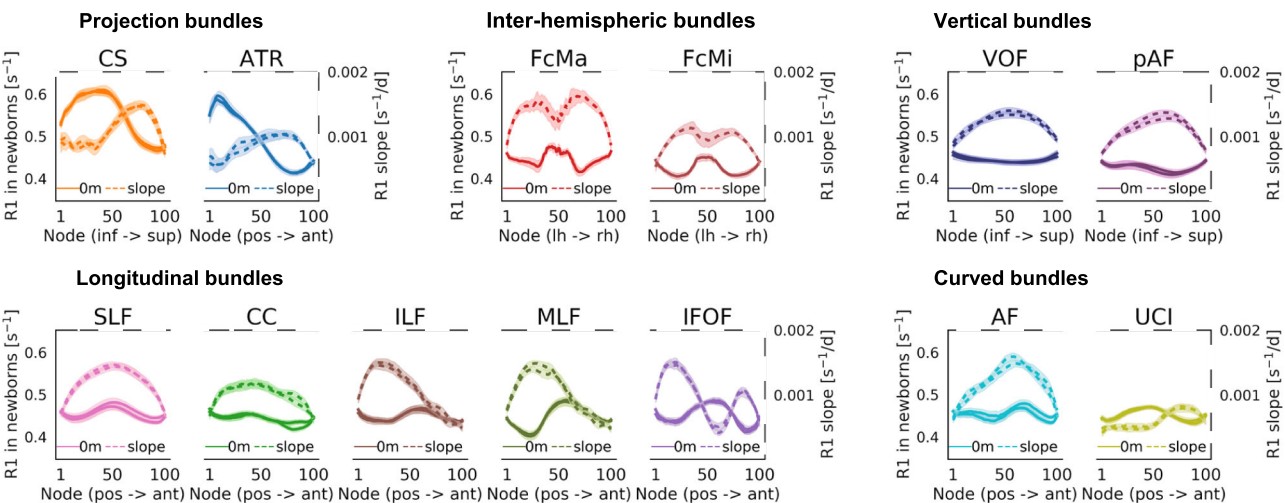

**Fig. 4 R1 development rate varies along the length of each bundle. a** Each panel jointly shows measured R1 in newborns (left *y*-axis, solid line) and the slope of R1 development (right *y*-axis, dashed line) at each node along the bundle. Higher R1 in newborns corresponds to higher values in solid lines. Faster development (more positive slope) corresponds to higher values in dashed lines. Lines from both hemispheres are presented separately but fall on top of each other. Shaded regions indicate standard error of measured R1 in newborns or slope of R1 development, respectively. Source data are provided as a Source Data file. CS cortico-spinal tract, ATR anterior thalamic radiation, FcMa forceps major, FcMi forceps minor, VOF vertical occipital fasciculus, pAF posterior arcuate fasciculus, AF arcuate fasciculus, UCI uncinate fasciculus, SLF superior longitudinal fasciculus, CC cingulum cingulate, ILF inferior longitudinal fasciculus, MLF middle longitudinal fasciculus, IFOF inferior frontal occipital fasciculus.

compared the slope to the measured R1 in newborns at each node (Fig. 4-solid lines). Results reveal that in all bundles there is a nonuniform rate of R1 increase along the length of the bundle. For example, the posterior ends of the inferior longitudinal fasciculus (ILF) and middle longitudinal fasciculus (MLF) show a larger change in R1 (more positive slope) than their anterior ends (Fig. 4). As R1 is linearly related to myelin fraction, these data suggest that myelination occurs at different rates along the length of these 24 bundles.

By plotting the rate of R1 development (slopes from LMMs; Fig. 4-dashed) along each bundle together with the measured R1 in newborns (Fig. 4-solid), we could also begin to assess the three developmental hypotheses. These plots revealed that in some bundles (e.g., the cortico-spinal tract (CS) or forceps (FcMa/ FcMi)) the rate of R1 increase is higher in locations along the bundle where R1 in newborns is lower. This suggests a negative relationship between R1 development and R1 at birth, consistent with the predictions of the speed-up hypothesis. In other bundles (e.g., posterior acuate fasciculus (pAF) or acuate fasciculus (AF)), R1 development rate varies substantially along the length of the bundle, but not in a clear relation to R1 measured in newborns. This is consistent with the predictions of the spatial gradient hypothesis. These qualitative observations provide the first evidence that multiple factors, including spatial gradients and R1 at birth, may contribute to the development of R1 along white matter bundles.

Like R1, MD shows (i) spatial variations along the length of each of the 24 bundles at all three time points, and (ii) significant development along the length of each bundle (Supplementary Fig. 6). Different than R1, (i) MD decreases with age (Supplementary Fig. 6), and (ii) the rate of MD development along the bundles shows a spatially distinct pattern compared to R1 (Supplementary Fig. 7). These analyses provide additional evidence that the development of MD in white matter bundles differs from R1 during early infancy.

**Spatial gradients and R1 at birth together explain R1 development.** The prior visualizations of R1 along white matter bundles suggest that both R1 at birth and the spatial location in the brain may contribute to the rate of R1 development during early infancy. To gain a global understanding of the spatial nature of R1 development across the white matter of the human brain, next, we visualized R1 measured in newborns and the rate of R1 development of white matter bundles in the 3D brain space of newborns (plotting every 10th node, Fig. 5), rather than along each individual bundle (as in Figs. 3 and 4). These 3D visualizations yield the following observations: (i) R1 in newborns varies spatially across the brain with overall highest values in central white matter and lowest values in frontal white matter (Fig. 5b), and (ii) the rate of R1 development varies spatially across the brain with faster increases in occipital and parietal white matter (yellow in Fig. 5c) and slower development in the temporal and frontal white matter (black in Fig. 5c). Overall, these visualizations suggest that both R1 at birth and spatial gradients across the brain appear to contribute to the rate of R1 development during early infancy. Thus, we next quantitatively tested the significance of each of these two factors separately, and then tested the viability of a model incorporating both factors. We applied a similar approach to MD (Fig. 5d, e).

First, we tested if the rate of R1 development is related to R1 measured in newborns (LMM relating R1 slope to R1 measured in newborns at every 10th node, with a random intercept per bundle). The speed-up hypothesis predicts a significant negative relationship but the starts-first/finishes-first hypothesis predicts a significant positive relationship. LMM results reveal a significant negative relationship between the rate of R1 development and R1 measured in newborns across the white matter ($\beta = -0.003$, $p < 0.0001$), that accounts for 40% of the variance in R1 slopes ($R^2 = 0.40$). That is, nodes that have higher R1 in newborns develop more slowly than nodes that have lower R1 in newborns, which is consistent with the speed-up hypothesis.

Second, we tested the spatial gradient hypothesis and evaluated if the rate of R1 development at each node is related to its spatial location in the brain (LMM relating R1 slope at every 10th node to the nodes average |x|, y, z coordinates in newborns, and their interactions |x|*y, |x|*z, and z*y; random intercept per bundle). Results show that there is a significant relationship between the rate of R1 development and spatial location along the z and y

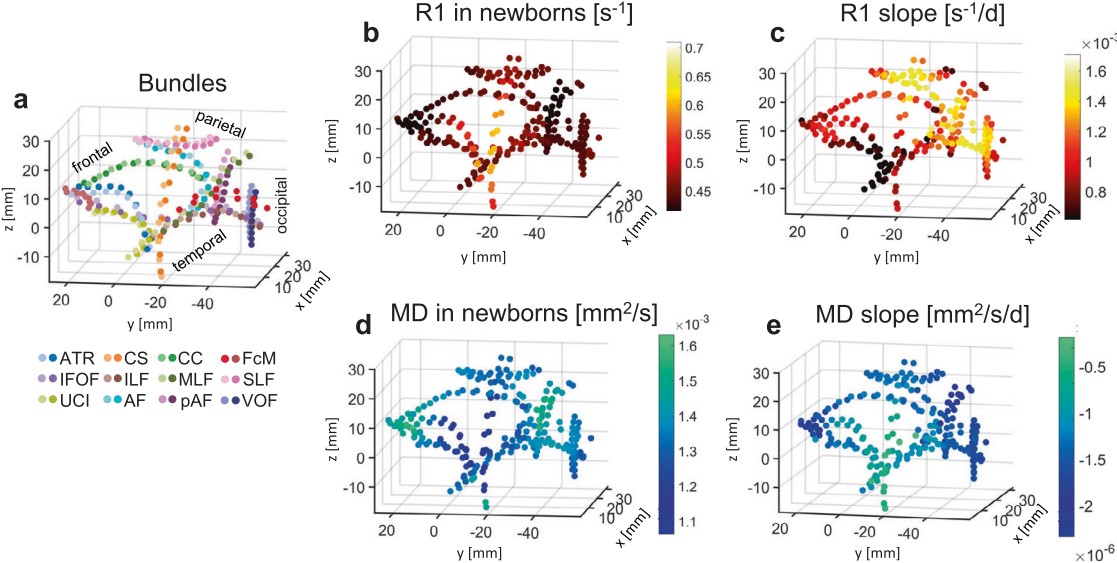

**Fig. 5 Spatial gradients and measurements at birth together explain R1 and MD development.** In all panels, each point is a node. In all plots only every 10th node of a bundle is plotted to ensure spatial independence of tested nodes. The coordinate of each node is the average |x|,y,z coordinate across newborns. As all data were acpc-ed, the 0,0,0 coordinate is the anterior commissure; |x|-axis is medial to lateral; y-axis is posterior to anterior; z-axis is inferior to superior. The axes are identical across panels. **a** 3D spatial layout of the 24 bundles in the average newborn brain volume. Nodes are color-coded by bundle (see legend, darker shades for left hemisphere); approximate lobe annotations are included to clarify the spatial layout. **b** 3D spatial layout of measured R1 at each node in newborns. Data are averaged across participants. Color indicates R1. **c** 3D spatial layout of R1 development rate (i.e., the slope estimated from LMM) at each node. **d** 3D spatial layout of measured MD at each node in newborns. Data are averaged across participants. Color indicates MD. **e** 3D spatial layout of MD development rate (i.e., the slope estimated from LMM) at each node. Source data are provided as a Source Data file. CS cortico-spinal tract, ATR anterior thalamic radiation, FcMa forceps major, FcMi forceps minor, VOF vertical occipital fasciculus, pAF posterior arcuate fasciculus, AF arcuate fasciculus, UCI uncinate fasciculus, SLF superior longitudinal fasciculus, CC cingulum cingulate, ILF inferior longitudinal fasciculus, MLF middle longitudinal fasciculus, IFOF inferior frontal occipital fasciculus.

axes and their combination (z: $\beta = 1.68 \times 10^{-4}$, $p < 0.0001$, y: $\beta = -1.10 \times 10^{-4}$, $p < 0.0001$, y*z: $\beta = 1.05 \times 10^{-4}$, $p < 0.0001$), and smaller but significant relationships along the |x| and |x|*y axes (x: $\beta = 4.19 \times 10^{-5}$, $p = 0.02$, |x|*y: $\beta = -4.74 \times 10^{-5}$, $p = 0.03$), which together explain 65% of the variance ($R^2 = 0.65$). These results support the spatial gradient hypothesis and suggest that development rate during infancy increases from inferior to superior, and from anterior to posterior, with additional gradients along medial to lateral directions.

As both R1 measured in newborns and spatial gradients explain a considerable amount of variance, a question remains if they are independent factors contributing to the rate of R1 development or not. Thus, we tested if the rate of R1 development at a node depends both on its spatial location and its R1 measured in newborns (LMM relating R1 slope at every 10th node to measured R1 in newborns and spatial coordinate: |x|, y, z, |x|*y, |x|*z, and z*y; with a random intercept per bundle). This combined model showed a significant negative relationship between the rate of R1 development and R1 measured in newborns: ($\beta = -0.001$; $p = 0.002$) and significant effects of spatial location along the z axis ($\beta = 1.53 \times 10^{-4}$, $p < 0.0001$), y-axis ($\beta = -1.11 \times 10^{-4}$, $p < 0.0001$), y*z axis ($\beta = 1.04 \times 10^{-4}$, $p < 0.0001$), and |x|*z axis ($\beta = 3.50 \times 10^{-5}$, $p = 0.03$). Overall, this combined model explains 67% of the variance in the rate of R1 development ($R^2 = 0.67$) and outperforms the best individual model, which was the spatial gradient model (likelihood ratio test, $p = 0.002$). Significant effects of R1 measured in newborns ($\beta = -0.0012$; $p = 0.006$) and spatial location (z axis: $\beta = 1.21 \times 10^{-4}$, $p < 0.0001$, y axis: $\beta = -1.19 \times 10^{-4}$, $p < 0.0001$, y*z axis: $\beta = 1.79 \times 10^{-4}$, $p < 0.0001$) were also observed when the first and last 10 nodes were excluded from the model, suggesting

that the observed effects are not predominantly driven by nodes in proximity of the cortical gray matter.

Similarly, we find that both MD measured in newborns and spatial gradients explain the rate of MD development in the white matter (Fig. 5d, e). As in the analyses of R1 development, we tested whether the rate of MD development across the white matter depends on both MD measured in newborns and spatial gradients using an LMM relating MD slope at every 10th node to measured MD in newborns and spatial coordinates (|x|, y, z, |x|*y, |x|*z, and z*y), with a random intercept per bundle. This combined model revealed a significant negative relationship between the rate of MD development and MD in newborns: ($\beta = -0.002$; $p < 0.0001$) as well as significant effects of spatial location along the x-axis ($\beta = -9.58 \times 10^{-8}$, $p = 0.0004$), the y-axis ($\beta = 9.78 \times 10^{-8}$, $p < 0.0001$), the z-axis ($\beta = -1.56 \times 10^{-7}$, $p < 0.0001$), and the |x| *y axis ($\beta = 6.41 \times 10^{-8}$, $p = 0.03$). Overall, this combined model explains 71% of the variance of the rate of MD development ($R^2 = 0.71$).

Together these analyses suggest that the nonuniform rates of R1 and MD development across the white matter during early infancy can be explained by two factors: initial values (measured in newborns) and spatial location in the brain (particularly along the inferior-to-superior and anterior-to-posterior axes).

## Discussion

By combining longitudinal measures of diffusion MRI and quantitative MRI with a novel approach for automated bundle quantification (babyAFQ) in individual infant's brains, we evaluated the longitudinal development of R1 and MD during early infancy along 24 white matter bundles, with three main

findings: First, in accordance with previous research[15], we find that across the white matter R1 systematically increases from newborns to 6-months-olds. Second, we find that the development of R1 is nonuniform across white matter bundles. Third, we discovered that the rate of R1 development during infancy is linked to both R1 at birth and spatial gradients. As R1 develops faster in sections of bundles that are less mature in newborns and as it is linearly related to myelin, these data support the speed-up hypothesis of infant myelin development. In addition, the rate of R1 development increases along the inferior-to-superior axis, the anterior-to-posterior axis, as well as along diagonal axes. These data suggest that myelination of the white matter during early infancy depends both on the initial myelin content at birth and spatial gradients.

Interestingly, the observed developmental pattern of MD showed both similarities and differences from the developmental pattern of R1. Consistent with the notion that increases in myelin (and R1) would be associated with decreases in MD, we find that MD in the white matter decreases during infancy, as reported previously[45–47]. However, we also find that the rate and pattern of MD and R1 development across the white matter are not identical. As MD is impacted by structural components of the white matter beyond myelin (e.g., fiber diameter and packing[18,23–25]) these differences (i) highlight the importance of using measures such as R1 which are linearly related to myelin[26,29–31] to assess myelin development specifically, and (ii) suggest that additional properties of white matter bundles beyond myelin are also developing during early infancy. Future histological measurements in postmortem pediatric samples may elucidate these mechanisms. In addition, while we have multiple measurements over time in the same individuals, our sample is limited to 13 infants and the first 6 months of life. Thus, it would be fruitful to extend these types of measurements to a larger sample of infants as well as over a longer period of infancy to better assess variability across individuals and determine the full developmental trajectory.

Crucially, as quantitative R1 measures are comparable across MRI scanners of the same field strength[9,15,26], we can compare our R1 measurements in infants to those of other populations. For example, we find that R1 in white matter bundles of full-term newborns ranges between 0.42–0.55 $[s^{-1}]$, which is higher than R1 in the white matter of preterm newborns, which ranges between 0.29–0.36 $[s^{-1}]$[48]. This observation suggests that at birth there is some level of myelin in all 24 bundles investigated here, contrasting with classic histological studies which reported myelin only in a handful of white matter bundles in newborns (e.g., the cortical-spinal tract)[2–5]. These contrasting results may be due to two reasons: On the one hand, as classic dissection studies used qualitative visual inspection of myelin stains in postmortem tissue, quantitative R1 measurements may simply be more sensitive to minimal amounts of myelin. On the other hand, more work is needed to elucidate what impacts R1 in the white matter bundles of the infant brain. While in the adult brain 90% of the variance in R1 in white matter bundles is related to myelin[29,31], in the sparsely myelinated infant brain, additional factors such as tissue density (e.g., proliferation of glia cells), water mobility, or changes in iron may contribute more strongly to R1.

Our measurements also reveal that R1 in 6-months-olds' bundles ranges between 0.54–0.73 $[s^{-1}]$, which is lower than the average R1 measured in adults' bundles, which ranges between 0.80–1.25 $[s^{-1}]$[44,49]. This comparison suggests that none of the 24 bundles investigated here are fully myelinated by 6 months of age. This is not surprising, as the average R1 across the white matter increases roughly linearly during the first year of life, after which its development slows down[15,50], but continues until early adulthood[44,51]. It is interesting that the bundles' R1 increases on average by ~17% (0.16 $[s^{-1}]$) within the first 6 months of life, as this change is larger than the increase of ~0.05 $[s^{-1}]$ observed over 10 years of childhood development[44] (from 8– to 18-years-of age). This observation highlights the profound changes occurring in the white matter during early infancy.

The finding that less mature white matter at birth myelinates faster during infancy has several implications. First, our data not only provides empirical evidence against the classic view that white matter develops in a strictly hierarchically manner from early sensory to higher-level cognitive regions[2,3], but also offers insights regarding the nature of white matter development in infancy. As myelination is experience-dependent[10–13], and we find that the rate of myelination after birth is negatively related to its initial (birth) level, one conjecture from our data is that the postnatal environment and experiences may produce a flurry of myelination during the first 6 months of life, overtaking earlier prenatal gradients. Second, as previous data has shown a link between cognitive development, processing speed, and myelin development during infancy and early childhood[52,53], we further hypothesize that the observed negative relationship between myelination at birth and the rate of myelin development is functionally relevant. For example, one consequence of this developmental trajectory is that it generates a more uniform distribution of myelin across the white matter, which may allow for more coordinated and efficient communication across the entire brain.

The rate of R1 development also varies spatially, with faster development occurring predominantly in the inferior-to-superior and anterior-to-posterior directions. As a result of these spatial gradients, the parietal and occipital lobe's white matter develops faster than central, frontal, and temporal white matter. This spatial pattern differs from observations made in preterm newborns before 40 weeks of gestation, which showed fastest development in the central white matter[48]. Instead, this pattern of R1 development during early infancy is more aligned with spatial gradients observed later in infancy and early childhood[15]. An open question is whether these spatial gradients are innate, or experience driven. One interesting avenue for answering this question in future research would be comparing the longitudinal development of spatial gradients across preterm newborns and full-term newborns. We hypothesize that the consequence of these spatial gradients may be to allow white matter that supports crucial functions such as vision (occipital lobe) and motor control (parietal lobe) to develop faster during infancy. Another interesting avenue for future studies could hence be to examine the relationship between R1 development in the white matter and R1 development in cortex[54,55].

Finally, our study may have important societal implications. First, as R1 values are quantitative and have units that can be numerically compared across scanners, populations, and individuals[26], our measurements in typically developing infants provide a key foundation for large-scale studies of infant brain development in typical[56,57] and clinical populations such as preterm infants[58], infants with cerebral palsy[59], or fetal alcohol spectrum disorders[60]. Second, our methodology is translatable to clinical settings as it is performed during natural sleep. Third, we developed an automated processing pipeline that simultaneously provides high throughput and high precision in individual infants. This level of precision may enable the early identification of developmental impairments in at-risk infants, which in turn may improve the efficacy of interventions[61]. Further, the spatial precision afforded by our methods may facilitate future work on the spatial dependency of both quantitative and diffusion metrics. For example, it would be interesting to formally assess if and how these measures change in spatial locations where multiple bundles cross each other.

In conclusion, we find that during early infancy myelin content at birth and spatial gradients of myelin development together explain the rate of myelin growth across the white matter of the human brain. This finding offers a parsimonious model of white matter development during early infancy. We hypothesize that this pattern of myelination during infancy enables some level of myelin becoming quickly available throughout the brain, to promote efficient and coordinated global communication, while at the same time prioritizing the development of most critical functions such as vision and motor coordination.

## Methods

**Participants**. Sixteen full-term and healthy infants (seven female) were recruited to participate in this study. Three infants provided no usable data because they could not stay asleep once the MRI sequences started and hence we report data from 13 infants (six female) across three timepoints: newborn ($N = 9$; age: 8–37 days), 3 months ($N = 10$; age: 79–106 days), and 6 months ($N = 10$; age: 167–195 days). Two participants were re-invited to complete scans for their 6-months session that could not be completed during the first try. Both rescans were performed within 7 days and participants were still within the age range for the 6-months timepoint. The participant population was racially and ethnically diverse reflecting the population of the Bay Area, including two Hispanic, nine Caucasian, two Asian, and three multiracial participants. Six out of the 13 infants participated in all three timepoints (0, 3, 6 months). Due to the Covid-19 pandemic and restricted research guidelines, data acquisition was halted. Consequently, the remaining infants participated in either 1 or 2 sessions.

Expectant mothers and their infants in our study were recruited from the San Francisco Bay Area using social media platforms. We performed a two-step screening process for expectant mothers. First, mothers were screened over the phone for eligibility based on exclusionary criteria designed to recruit a sample of typically developing infants and second, eligible expectant mothers were screened once again after giving birth. Exclusionary criteria for expectant mothers were as follows: recreational drug use during pregnancy, significant alcohol use during pregnancy (more than three instances of alcohol consumption per trimester; more than 1 drink per occasion), lifetime diagnosis of autism spectrum disorder or a disorder involving psychosis or mania, taking prescription medications for any of these disorders during pregnancy, insufficient written and spoken English ability to understand the instructions of the study, or learning disabilities that would preclude participation in the study. Exclusionary criteria for infants were: preterm birth (<37 gestational weeks), low birthweight (<5 lbs 8 oz), small height (<18 inches), any congenital, genetic, and neurological disorders, visual problems, complications during birth that involved the infant (e.g., NICU stay), history of head trauma, and contraindications for MRI (e.g., metal implants). Study protocols for these scans were approved by the Stanford University Internal Review Board on Human Subjects Research. Participants were compensated with 25 dollars per hour for their participation in the study.

**Data acquisition procedure**. Data collection procedure was developed in a recent study[54]. All included participants completed the multiple scanning protocols needed to obtain anatomical MRI, qMRI, and dMRI data. Data were acquired at two identical 3 T GE Discovery MR750 Scanners (GE Healthcare) with Nova 32-channel head coils (Nova Medical) located at Stanford University: (i) Center for Cognitive and Neurobiological Imaging (CNI) and (ii) Lucas Imaging Center. As infants have low weight, all imaging was done with first level SAR to ensure their safety.

Scanning sessions were scheduled in the evenings close in time to the infants' typical bedtime. Each session lasted between 2.5 and 5 h including time to prepare the infant and waiting time for them to fall asleep. Upon arrival, caregivers provided written, informed consent for themselves and their infant to participate in the study. Before entering the MRI suite, both caregiver and infant were checked to ensure that they were metal-free, and caregivers changed the infant into MR-safe cotton onesies and footed pants provided by the researchers. The infant was swaddled with a blanket with their hands to their sides to avoid their hands creating a loop. During sessions involving newborn infants, an MR-safe plastic immobilizer (MedVac, www.supertechx-ray.com) was used to stabilize the infant and their head position. Once the infant was ready for scanning, the caregiver and infant entered the MR suite. The caregiver was instructed to follow their child's typical sleep routine. As the infant was falling asleep, researchers inserted soft wax earplugs into the infant's ears. Once the infant was asleep, the caregiver was instructed to gently place the infant on a makeshift cradle on the scanner bed, created by weighted bags placed at the edges of the bed to prevent any side-to-side movement. Finally, to lower sound transmission, MRI compatible neonatal Noise Attenuators (https://newborncare.natus.com/products-services/newborn-care-products/nursery-essentials/minimuffs-neonatal-noise-attenuators) were placed on the infant's ears and additional pads were also placed around the infant's head to stabilize head position.

An experimenter stayed inside the MR suite with the infant during the entire scan. For additional monitoring of the infant's safety and tracking of the infant's head motion, an infrared camera was affixed to the head coil and positioned for viewing the infant's face in the scanner. The researcher operating the scanner monitored the infant via the camera feed, which allowed for the scan to be stopped immediately if the infant showed signs of waking or distress. This setup also allowed tracking the infant's motion; scans were stopped and repeated if there was excessive head motion. To ensure scan data quality, in addition to real-time monitoring of the infant's motion via an infrared camera, MR brain image quality was also assessed immediately after acquisition of each sequence and sequences were repeated if necessary.

**Data acquisition parameters and preprocessing**

*Anatomical MRI*. T2-weighted images were acquired and used for tissue segmentations. T2-weighed image acquisition parameters: TE = 124 ms; TR = 3650 ms; echo train length = 120; voxel size = 0.8 mm³; FOV = 20.5 cm; Scan time: 4 min and 5 s.

We generated gray/white matter tissue segmentation of all infants and timepoints and used these segmentations to optimize tractography (anatomically constrained tractography, ACT[62]). The T2-weighted anatomy, and a synthetic T1-weighted brain image generated from the SPGRs and IR-EPI scans using mrQ software (https://github.com/mezera/mrQ) were aligned and used for segmentation. Multiple steps were applied to generate accurate segmentation of each infant's brain at each timepoint[54]. (1) An initial segmentation of gray and white matter was generated from the T1-weighted brain volume using infant FreeSurfer's automatic segmentation code (infant-recon-all; https://surfer.nmr.mgh.harvard.edu/fswiki/infantFS[63]). (2) A second segmentation was done using the T2-weighted anatomical images, which have a better contrast between gray and white matter in young infants, using the brain extraction toolbox (Brain Extraction and Analysis Toolbox, iBEAT, v-2.0 cloud processing, https://ibeat.wildapricot.org/[64–66]). (3) The iBEAT segmentation, which was more accurate, was manually corrected to fix segmentation errors (such as holes and handles) using ITK-SNAP (http://www.itksnap.org/). (4) The iBEAT segmentation was then reinstalled into FreeSurfer and the resulting segmentation in typical FreeSurfer format was used to optimize tractography. We also identified the ventricles in each infant using the iBEAT ventricle labels. We visually inspected these labels in each infant and time point and manually edited them where necessary, to ensure that all ventricle voxels were included in the label. We then used this label as a mask, thus removing the ventricles from the R1 and MD maps, to limit the impact of partial volume artifacts between cerebral spinal fluid and white matter in neighboring bundles.

*Quantitative MRI*. An inversion-recovery EPI (IR-EPI) sequence was used to estimate relaxation time (R1) at each voxel. Spoiled-gradient echo images (SPGRs) were used together with the EPI sequence to generate whole-brain synthetic T1-weighted images. We acquired 4 SPGRs whole-brain images with different flip angles: $\alpha = 4°, 10°, 15°, 20°$; TE = 3 ms; TR = 14 ms; voxel size = 1 mm³; number of slices = 120; FOV = 22.4 cm; Scan time: 4 times ~5 min. We also acquired multiple inversion times (TI) in the IR-EPI using a slice-shuffling technique[67]: 20 TIs with the first TI = 50 ms and TI interval = 150 ms as well as a second IR-EPI with reverse-phase encoding direction. Other acquisition parameters were: voxel size = 2 mm³; number of slices = 60; FOV = 20 cm; in-plane/through-plane acceleration = 1/3; Scan time = two times 1:45 min.

IR-EPI data were used to estimate R1 (R1 = 1/T1) in each voxel. First, as part of the preprocessing, we performed susceptibility-induced distortion correction on the IR-EPI images using FSL's top-up and the IR-EPI acquisition with reverse-phase encoding direction. We then used the distortion corrected images to fit the T1 relaxation signal model using a multi-dimensional Levenberg-Marquardt algorithm[68]. The signal equation of T1 relaxation of an inversion-recovery sequence is an exponential decay: $S(t) = a(1 - be^{-t/T1})$, where $t$ is the inversion time, $a$ is proportional to the initial magnetization of the voxel, $b$ is the effective inversion coefficient of the voxel (for perfect inversion $b = 2$). We applied an absolute value operation on both sides of the equation and used the resulting equation as the fitting model. We use the absolute value of the signal equation because we use the magnitude images to fit the model. The magnitude images only keep the information about the strength of the signal but not the phase or the sign of the signal. The output of the algorithm is the estimated T1 in each voxel. From the T1 estimate, we calculated R1 (R1 = 1/T1) at each voxel.

*Diffusion MRI*. We obtained dMRI data with the following parameters: multi-shell, #diffusion directions/b-value = 9/0, 30/700, 64/2000; TE = 75.7 ms; TR = 2800 ms; voxel size = 2 mm³; number of slices = 60; FOV = 20 cm; in-plane/through-plane acceleration = 1/3; scan time: 5:08 min. We also acquired a short dMRI scan with reverse phase encoding direction and only 6 b = 0 images (scan time 0:20 min).

DMRI preprocessing and tractography were performed in accordance with recent work from the developing human connectome project[69,70], using a combination of tools from MRtrix3[71,72] (github.com/MRtrix3/mrtrix3) and mrDiffusion (http://github.com/vistalab/vistasoft). We (i) denoised the data using a principal component analysis[73], (ii) used FSL's top-up tool (https://fsl.fmrib.ox.ac.uk/) and one image collected in the opposite phase-encoding direction to correct for susceptibility-induced distortions, (iii) used FSL's eddy to perform eddy current and motion correction, whereby motion correction included

outlier slice detection and replacement[74], and (iv) performed bias correction using ANTs[75]. The preprocessed dMRI images were registered to the whole-brain T2-weighted anatomy using whole-brain rigid-body registration and alignment quality was checked for all images. dMRI quality assurance was also performed. Across all acquisitions, <5% ± 0.72% of dMRI images were identified as outliers by FSL's eddy tool. We found no significant effect of age across the outliers (no main effect of age: $F(2,26) = 1.97$, $p = 0.16$, newborn: $1.07 + 0.88\%$; 3 months: $0.4 + 0.40\%$; 6 months: $0.67 + 0.85\%$), suggesting that the developmental data was well controlled across all time-points.

Next, voxel-wise fiber orientation distributions (FODs) were calculated using constrained spherical deconvolution (CSD) in MRtrix3[71] (Supplementary Fig. 2). We used the Dhollander algorithm[76] to estimate the three-tissue response function, and we lowered the FA threshold to 0.1 to account for the generally lower FA in infant brains. We computed FODs with multi-shell multi-tissue CSD[77] separately for the white matter and the CSF. As in previous work[69], the gray matter was not modeled separately, as white and gray matter do not have sufficiently distinct b-value dependencies to allow for a clean separation of the signals. Finally, we performed multi-tissue informed log-domain intensity normalization.

We used MRtrix3[71] to generate a whole-brain white matter connectome for each infant and time point. Tractography was optimized using the tissue segmentation from the anatomical MRI data (anatomically constrained tractography, ACT[62]). We argue that this approach is particularly useful for infant data, as gray and white matter cannot be separated in the FODs. For each connectome, we used probabilistic fiber tracking with the following parameters: algorithm: IFOD1, step size: 0.2 mm, minimum length: 4 mm, maximum length: 200 mm, maximum angle: 15°. Seeds for tractography were randomly placed within the gray/white matter interface (from anatomical tissue segmentation), which enabled us to ensure that tracts reach the gray matter. Each connectome consisted of 2 million streamlines. MRtrix3 software was also used to fit tensor kurtosis models from which we estimated MD maps for each individual.

**Bundle delineation with baby automated fiber quantification (babyAFQ).** Here we developed a new toolbox (babyAFQ) that identifies white matter bundles in individual infants. BabyAFQ is openly available as a novel component of AFQ[32] (https://github.com/yeatmanlab/AFQ/tree/master/babyAFQ) and identifies the following bundles in infants (Fig. 1): anterior thalamic radiation (ATR), cortico-spinal tract (CS), posterior arcuate fasciculus (pAF), vertical occipital fasciculus (VOF), forceps major (FcMa), forceps minor (FcMi), arcuate fasciculus (AF), uncinate fasciculus (UCI), superior longitudinal fasciculus (SLF), cingulum cingulate (CC), inferior longitudinal fasciculus (ILF), inferior frontal occipital fasciculus (IFOF) and the middle longitudinal fasciculus (MLF).

BabyAFQ uses anatomical ROIs as waypoints for each bundle. That is, a given tract is considered a candidate for belonging to a bundle only if it passes through all waypoints associated with that bundle. The waypoint ROIs were adjusted from those commonly used in adults[36] to better match the head size and white matter organization of infants (Supplementary Fig. 3). Specifically, we: (i) spatially restricted some of the waypoint ROIs to account for the more compact infant brain, (ii) introduced a third waypoint for curvy bundles, (iii) generated new volumetric waypoint ROIs for the VOF (Supplementary Fig. 4) to allow identification of the VOF in brains for which cortical surface reconstructions are not available, and (iv) added new waypoint ROIs for identifying the MLF, as the MLF was not included in prior AFQ versions. Critically, these waypoints were defined in a neonate infant template brain (UNC Neonatal template[35]) and are transformed from this template space to each individual infant's brain space before bundle delineation. The use of an infant template brain is critical as commonly used adult templates, such as the MNI brain, are substantially larger and difficult to align to infants' brains. In cases where a given tract is a candidate for multiple bundles, a probabilistic atlas, which is also transformed from the infant template space to the individual infant brain space, is used to determine which bundle is the better match for the tract. Bundles are then cleaned by removing tracts that exceed a gaussian distance of 4 standard deviations from the core of the bundle. Critically, babyAFQ was designed to seamlessly integrate with AFQ, so that additional tools for plotting, tract profile evaluation, and statistical analysis can be applied after bundle delineation.

**BabyAFQ quality assurance.** To evaluate the quality of the bundle delineation by babyAFQ, we compared the automatically identified bundles to manually delineated gold-standard bundles. Manual bundle delineation was performed for the newborns in DSI Studio (http://dsi-studio.labsolver.org/) by two anatomical experts who were blind to the results of babyAFQ. As a benchmark, we also delineated bundles with AFQ, which was developed using adult data, and compared these bundles to the gold-standard bundles. For both babyAFQ and AFQ we quantified the spatial overlap between the automatically identified bundles and the manually identified bundles using the dice coefficient[78] (DC): $DC = \frac{2|A \cap B|}{|A| + |B|}$, where $|A|$ are voxels of automatically identified bundles, $|B|$ are voxels of the manual bundles, and $|A \cap B|$ is the intersection between these two sets of voxels (Fig. 1b). We compared dice coefficients between babyAFQ and AFQ in two repeated measures analyses of variance (rmANOVAs). First, a two-way rmANOVA with AFQ-type and bundle as factors allowed us to evaluate the effect of AFQ type

across all bundles. Second, a three-way rmANOVA on bilateral bundles with AFQ-type, bundle, and hemisphere as factors, enabled us to test for additional hemispheric differences. Finally, we also used the dice coefficients to test if tracts identified as belonging to the VOF were similar or different across methods—using volumetric way-point ROIs vs. surface ROIs (Supplementary Fig. 4).

In addition to the quantitative evaluation, we examined all bundles delineated using babyAFQ and AFQ qualitatively at all time-points (Supplementary Fig. 8) to evaluate how well they match the typical spatial extent and trajectory across the brain. We also created, with pyAFQ[34], an interactive 3D visualization of an example infant's bundles at each time point: 0 months, 3 months, and 6 months.

**Modeling R1 development.** After identifying all bundles with babyAFQ, we modeled their R1 development using LMMs. First, we modeled mean R1 development within each bundle using LMMs with age as predictor and a random intercept (estimated R1 at birth) for each individual (Fig. 2a). We used model comparison (likelihood ratio tests) to determine that LMMs allowing different slopes for each individual do not better explain the data compared to LMMs using a single slope across individuals. To evaluate differences in developmental trajectories between bundles, we plotted the mean R1 measured in newborns (Fig. 2b) and well as the mean R1 development rate (slopes of LMMs) for each bundle (Fig. 2c).

Next, we evaluated the development of R1 along the length of each bundle. For this, we divided each bundle into 100 equidistant nodes and evaluated R1 at each time-point in each node (Fig. 3). We then determined the rate of R1 development at each node (one LMM per node; random intercepts for each individual as above). For each bundle, we then plotted R1 measured in newborns and the rate of R1 development across nodes to visualize their relationship along each bundle (Fig. 4).

Finally, we evaluated the relationship between the rate of R1 development (LMM slope) and both the measured R1 in newborns as well as the spatial location in the brain (Fig. 5). This analysis was done for every 10th node along each bundle to ensure spatial independence across nodes within a bundle. All subplots in Fig. 5 show the data at each node plotted at their average location in the newborn's brain (average$|x|$, y and z coordinates in the newborn sample). For the x axis we used the $|x|$ coordinates, as previous work suggests a medial to lateral spatial gradient of development across both hemispheres of the infant brain[5]. As all newborn brain volumes were rotated to be aligned to a plane crossing through the anterior and posterior commissures (i.e., brain volumes were acpc-ed), the (0,0,0) coordinate corresponds to the average coordinate of the anterior commissure across newborns. Figure 5a is included to orient the reader to the spatial layout in these plots. Figure 5b shows the spatial layout of measured R1 in newborns across the white matter, and Fig. 5c shows the spatial layout of R1 development rate across the white matter.

We quantified the relationship between R1 development rate and initial R1 as well as spatial location via a series of LMMs. In the first LMM, we related R1 development rate to R1 measured in newborns, with a random intercept for each bundle:

(1)   R1Slope ~ 1 + R1 in Newborns + (1|Bundle).
       In the second LMM, we related R1 development rate to location in the brain ($|x|$, y, z, $|x|*y$, $y*z$, and $z*|x|$ coordinates, all coordinates were z-scored before including interaction terms), with a random intercept per bundle:
(2)   R1Slope ~ 1 + $|x|$ + y + z + $|x|*y$ + $|x|*z$ + $y*z$ + (1|Bundle).
       In the third model, we related R1 development to both R1 measured in newborns as well as spatial location, with a random intercept per bundle:
(3)   R1Slope ~ 1 + R1 in Newborns + $|x|$ + y + z + $|x|*y$ + $|x|*z$ + $y*z$ + (1|Bundle).

We used a likelihood ratio test to assess whether this third model outperforms the second model. Similar analyses were also performed on MD data, to relate our findings to previous work. MD results are presented in Figs. 2d, e, 5d, e and Supplementary Figs. 5–7.

**Reporting summary.** Further information on research design is available in the Nature Research Reporting Summary linked to this article.

## Data availability
All data required to generate the main figures are provided as a Source Data file with this paper and is also made available in GitHub (https://github.com/VPNL/babyWmDev) and on Zenodo (https://doi.org/10.5281/zenodo.5788646). Source data are provided with this paper.

## Code availability
The data were analyzed using open source software, including mrDiffusion and MRtrix3[71]. We developed a new toolbox for automated fiber quantification in individual infants (babyAFQ) and make it openly available (https://github.com/yeatmanlab/AFQ/tree/master/babyAFQ, for an example of how to run babyAFQ see https://figshare.com/s/456282406044bbb490ee). Code to reproduce all figures is available in GitHub (https://github.com/VPNL/babyWmDev) and on Zenodo (https://doi.org/10.5281/zenodo.5788646).

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

## Acknowledgements

The research was funded by: Wu Tsai Neurosciences Institute Big Idea Neurodevelopment Grant awarded to K.G.S., R21 EY030588 grant awarded to K.G.S., and the Center for Mind, Brain and Behavior – CMBB, Philipps-Universität Marburg, and Justus-Liebig-Universität Giessen awarded to M.G. We would like to thank all participating families, as well as KK Barrows, Amy Kang, Javier Lopez, Laura Villalobos, Nancy Lopez-Alvarez, and Lois Williams for their help segmenting the infant brains into white and gray matter. We would also like to thank Jiyeong Ha for her contributions towards data quality assurance and Caitlyn Estrada for her contribution to data collection.

## Author contributions

M.R., H.K., F.R.Q., and M.G. collected the data. M.R., V.N., H.K., and F.R.Q. generated gray/white matter segmentations and R1 maps. H.W. developed scanning sequences. M.G. and J.D.Y. developed babyAFQ and data analysis pipeline. M.G., J.D.Y., and K.G.S. analyzed data. M.G. and K.G.S. wrote the manuscript. All authors read and edited the manuscript.

## Funding

## Competing interests

The authors declare no competing interests.
