## [Peer Review File · Nature Communications]

White matter myelination during early infancy is linked to spatial gradients and myelin content at birthReviewers' comments:

Reviewer #1 (Remarks to the Author):

The paper "Catch me if you can: Least myelinated white matter develops fastest during early infancy" is a well-written manuscript that addresses questions that I believe are extremely relevant to understand brain maturation. When looking at the similar literature I would be tempted to say that the combination of DTI and subject specific bundle definition is superior to similar attempts when studying maturation using T1 relaxometry or mcDESPOT. Nevertheless, I am under the impression that the novelty of the methods and the impact of the results are overemphasized and should be tuned down.

The paper states that it wants to answer a set of hypothesis and starts with the following predictions:

- (i) bundles that are more myelinated at birth, will have lower T1 in newborns than less myelinated bundles,
- (ii) if myelin increases from 0 to 6 months, then T1 will decrease from 0 to 6 months, and
- (iii) if T1 development follows the starts-first/finishes first hypothesis T1 will decrease faster in bundles with lower T1 at birth, but if T1 development follows the catch-up hypothesis T1 will decrease faster in bundles with higher T1 at birth.

I believe the first two are not exactly predictions but facts well described in literature (some of it cited in the manuscript and other eventually missing –see further literature at the end of my report. The T1 is well known to be longer at birth and to shorten during development. T1 in the brain is widely linked to myelin concentration but this not actually addressed in this paper, rather it is assumed.

The last prediction has some physical limitations that would require reanalyzing the data. As the approach used makes it likely to find the predicted result.

1. When linking relaxation times to a source of contrast, the correct formalism is to use relaxation rates and relaxivities. In that scenario

$$R_{1,0} = 1/T_{1,0} = R_{1,0} + r_{1,myelin} [Myelin]$$

Where $r_{1,myelin}$ is the relaxivity of the contrast source (myelin in this simplified expression), and $[Myelin]$ would be the concentration of myelin.

As a result, it is $\Delta R_{1,0}$ that could be directly linked to myelination (when we are not considering multiple compartments and no exchange between compartments). It also results from this equation that a given increase of myelin, ΔM , would result in a larger ΔT_1 if the fiber would initially be less myelinated. Analytically, it can be found that $\Delta T_{1,0}$ and ΔM are related by:

$$\Delta T_{1,0} = r_{1,myelin} \Delta M T_{1,0}^2 / (1 - r_{1,myelin} \Delta M T_{1,0})$$

Where $T_{1,0}$ is the initial $T_{1,0}$ of a given fiber bundle. As we are working in a regime where $r_{1,myelin} \Delta M < 1$ (in this paper the R_1 varied from 0,3 to 0,5), an increase in myelin will have a bigger effect in T_1 of tissues with a larger initial T_1 .

2 Another of the findings reported in the paper is that at 6 months myelination is not complete. I don't have the impression that this is a new or surprising finding as can be seen in some of the covered literature. Indeed there is literature from some of the authors of this paper showing myelination happening until the late 30's for various white matter tracts.

3 There is some related work that should be cited:

Soun JE, Liu MZ, Cauley KA, Grinband J. Evaluation of neonatal brain myelination using the T1- and T2-weighted MRI ratio. *Journal of Magnetic Resonance Imaging*. 2017;46(3):690–6.

Eminian S, Hajdu SD, Meuli RA, Maeder P, Hagmann P. Rapid high resolution T1 mapping as a marker of brain development: Normative ranges in key regions of interest. *PLoS One*, 2018; 13(6) Available from: <https://www.ncbi.nlm.nih.gov/pmc/articles/PMC6002025/>

Also the findings in these other papers:

Deoni SCL, Dean DC, O'Muircheartaigh J, Dirks H, Jerskey BA. Investigating white matter development in infancy and early childhood using myelin water fraction and relaxation time mapping. *NeuroImage*. 2012 Nov 15;63(3):1038–53.

Schneider J, Kober T, Graz MB, Meuli R, Hüppi PS, Hagmann P, et al. Evolution of T1 Relaxation, ADC, and Fractional Anisotropy during Early Brain Maturation: A Serial Imaging Study on Preterm Infants. *American Journal of Neuroradiology*. 2016 Jan 1;37(1):155–62.

Should be compared to the current findings to see if the question asked here was not already virtually answered in previous literature.

Minor points

Line 313 It is mentioned that SPGR data is acquired together with the Inversion recovery to compute the relaxation times, but the methods only mention the use of the inversion recovery sequence.

The novelty of the methods used for defining the fiber bundles are relatively standard and should therefore be tuned down.

Reviewer #2 (Remarks to the Author):

This technically impressive paper looks at the rate of myelination of white matter in the first 6 postnatal months using a longitudinal design. The authors segregate fibre bundles according to whether they are association, projection or callosal using an automated pipeline and use T1 relaxation time as a proxy of myelin content.

The MRI methods employed are very impressive, the resulting tractography in individual babies is excellent. This is a very difficult feat in 3/6 month olds. The tractometry approach in figure 4 is really interesting and tells a lot about the tracks change over time. It would be really interesting to see what the slopes are at the cross sections of overlapping tracks - e.g. the cortico spinal track has a higher rate of change in superior regions where it intersects with other white matter bundles.

My main criticism is with the central claim of the paper, that least myelinated white matter develops fastest in a catch up way. I have no problem as to whether it does or not, it's just that the age range sampled (0-6 months) and the size of the sample (only 6 with all three timepoints and some with only 1 timepoint from what I can read in the methods) make it impossible to say this.

Earlier work looking at relaxometry mapping with T1 would indicate effectively linear growth of R1 until ~18 months (Deoni et al 2012, *Neuroimage*) and I feel you need to follow to at least 12 to support the title of the paper. Moreover, from the authors plots in Figure 2, the corticospinal track still has lower T1 than all other bundles at 6 months, reinforcing that fast developing tracts haven't caught up yet.

I appreciate the argument that standard deviation across T1 of the tracks is reducing as you move up in age but as this is reflected in a more flat contrast on a T1 generally. The correlation in Figure 2b is also partly driven by projection bundles having such markedly distinct T1 at birth, in their absence I'm not sure the correlation would be there.

The discussion mentions that the data will help interpret developmental trajectories of diffusion metrics - this data is here why not show it?

They also mention that rate of change may be functionally relevant - there are a few papers from the Brown group that look specifically at rate of change and cognitive ability. It's a different measure of myelin but is strongly related to T1. According to those papers, rate of myelination is functionally relevant.

We would like to thank both reviewers for providing constructive feedback that has enabled us to improve our manuscript. Additionally, we thank the reviewers for highlighting the importance of our work and underscoring the technical quality of our research. Here, we provide a detailed point-by-point response to each of your comments (reviewers comments in black, our response in green). We begin by summarizing the key changes that we implemented in this revision:

- 1) In response to Reviewer 1's main concern, we have implemented a simulation to test if we can use T1 and/or R1 to test the developmental hypotheses (new **Supplementary Figure 1**). Based on this simulation, we concur with the reviewer that because R1 (but not T1) shows a linear relationship with myelin, only R1 can be used as a metric to disambiguate the developmental hypotheses. We thank the reviewer for pointing this out, as we had not considered this implication before. Consequently, we now report R1 rather than T1 throughout the manuscript (new **Figures 2-4**).
- 2) In response to Reviewer 2's comment 1, we now refer to the hypothesis suggesting a negative correlation between the rate of myelin development and myelin content at birth as the *speed-up hypothesis* rather than the *catch-up hypothesis*. While the term *catch-up* has been used in other developmental studies, we agree with Reviewer 2 that since we only measure brain development during the first 6 months of live, our data does not enable us to assess at what time bundles "catch-up" with each other. In contrast, the *speed-up hypothesis* is directly supported by our data.
- 3) In response to Reviewer 2's comment 3, we added new data on the longitudinal development of mean diffusivity (MD) during early infancy (new **Supplementary Figures 5-8**).
- 4) In response to comments raised by both reviewers we have expanded and reframed the introduction and discussion, added additional references, and toned-down descriptions of novelty. This allowed us to better situate our findings relative to prior research and distill the advancements of the present study.
- 5) Finally, in addition to testing the speed-up, and starts-first/finishes-first hypotheses, we now also tested a third development hypothesis, the spatial gradient hypothesis. This new hypothesis is based on literature highlighted by Reviewer 1, that suggests that white matter myelination during infancy may progress in a spatially organized manner that is unrelated to myelin content at birth. In new **Figure 5** and the accompanying Results section, we test the viability of several quantitative models of R1 development. We find that the best model for the rate of R1 development in early infancy has significant contributions of both R1 at birth and spatial gradients. The former showing a negative contribution, consistent with the speed-up hypothesis, and the latter showing increased development rate in the inferior-to-superior and anterior-to-posterior directions. Together these new results provide a parsimonious explanation of myelin development across the white matter of the human brain during early infancy.

Point by point responses to Reviewers' comments:

Reviewer #1 (Remarks to the Author):

The paper "Catch me if you can: Least myelinated white matter develops fastest during early infancy" is a well-written manuscript that addresses questions that I believe are extremely relevant to understand brain maturation. When looking at the similar literature I would be tempted to say that the combination of DTI and subject specific bundle definition is superior to similar attempts when studying maturation using T1 relaxometry or mcDESPOT. Nevertheless, I am under the impression that the novelty of the methods and the impact of the results are overemphasized and should be tuned down.

Thank you for your feedback and thoughtful review. As detailed below, we have toned down descriptions of novelty as you suggested. Nonetheless, we also wanted to take the opportunity to briefly highlight the novelty of the current study: While we agree that since the classical work of Flechsig (1921) it is known that white matter myelinates from infancy onwards, what is highly debated till the present is how white matter myelinates and in what sequence. What we aimed to accomplish in this study is resolving this gap in knowledge. This advancement is significant and was accomplished by using a novel combination of methods including longitudinal measurements of dMRI and quantitative R1 in infants during the first 6 months of life. By using a quantitative metric (relaxation time (R1)) that is directly related to myelin fraction in a voxel, the current study was able to quantitatively distinguish among development hypotheses. Finally, we developed a new analysis tool optimized for infants (babyAFQ) to automatically identify white matter bundles in each infant's brain using dMRI data. This new tool improves the precision of analyses, as it enables measurements in the native infant's brain space, and along each bundle.

The paper states that it wants to answer a set of hypothesis and starts with the following predictions:
 (i) bundles that are more myelinated at birth, will have lower T1 in newborns than less myelinated bundles,
 (ii) if myelin increases from 0 to 6 months, then T1 will decrease from 0 to 6 months, and
 (iii) if T1 development follows the starts-first/finishes first hypothesis T1 will decrease faster in bundles with lower T1 at birth, but if T1 development follows the catch-up hypothesis T1 will decrease faster in bundles with higher T1 at birth.

I believe the first two are not exactly predictions but facts well described in literature (some of it cited in the manuscript and other eventually missing –see further literature at the end of my report. The T1 is well known to be longer at birth and to shorten during development. T1 in the brain is widely linked to myelin concentration but this not actually addressed in this paper, rather it is assumed.

Thank you for your summary and comment. To address this concern, we have (i) changed the wording to clarify what is expected based on the current literature and what are our predictions throughout the manuscript and (ii) cited additional references, including those you list below. We acknowledge that a few references were missing because of strict limitations on the number of references allowed by the journal, but nevertheless, in the revision, we were able to add these references.

We have rewritten the paragraph regarding the predictions referred above. On pages 4-5, we now write:
“As increases in myelin in the white matter generate linear increases in R1, the developmental hypotheses tested here make the following predictions: The starts-first/finishes-first hypothesis predicts that during the first 6 months of life, R1 will increase faster in white matter that is more myelinated at birth and hence has higher R1 values in newborns. The speed-up hypothesis predicts the opposite, that during the first 6 months of life, R1 will increase faster in white matter that has lower R1 values in newborns. Finally, the spatial gradient hypothesis predicts spatial differences in the development of R1 across the white matter, that cannot be explained by differences in R1 values in newborns.”

The last prediction has some physical limitations that would require reanalyzing the data. As the approach used makes it likely to find the predicted result.

1)When linking relaxation times to a source of contrast, the correct formalism is to use relaxation rates and relaxivities. In that scenario

$$R1 = 1/T1 = R1,0 + r1\text{myelin}[\text{Myelin}]$$

Where R1 is the relaxivity of the contrast source (myelin in this simplified expression), and [Myelin] would be the concentration of myelin.

As a result, it is $\Delta R1$ that could be directly linked to myelination (when we are not considering multiple compartments and no exchange between compartments). It also results from this equation that a given increase of myelin, ΔM , would result in a larger $\Delta T1$ if the fiber would initially be less myelinated. Analytically, it can be found that $\Delta T1$ and ΔM are related by:

$$\Delta T1 = r1 \Delta M T1,i^2 / (1 - r1 \Delta M T1,i)$$

Where $T_{1,i}$ is the initial T_1 of a given fiber bundle. As we are working in a regime where $r_1 \Delta M < 1$ (in this paper the R_1 varied from 0,3 to 0,5), an increase in myelin will have a bigger effect in T_1 of tissues with a larger initial T_1 .

Thank you for pointing out this important concern. We hadn't considered the implication of R_1 varying linearly with myelin and T_1 varying with $1/\text{myelin}$. To better understand the issue, we first ran a simulation (new **Supplementary Fig. 1**, also shown below) to model the relationships between a change in myelin (Δmyelin , dm) and a change in T_1 (ΔT_1 or dT_1 in the figure) or R_1 (ΔR_1 or dR_1 in the figure), including testing experimental predictions based on the developmental hypotheses.

The simulation confirmed the reviewer's concern: because T_1 varies with $1/\text{myelin}$ (new **Supplementary Fig. 1a, top left**) the same unit change in myelin will have a bigger effect in T_1 of tissue with a lower initial myelin fraction (new **Supplementary Fig. 1a, top right**). Thus, because a change in T_1 depends both on myelin at birth and the rate of myelin change, we cannot distinguish developmental hypotheses by plotting the change in T_1 (dT_1) vs T_1 at birth. As illustrated in the bottom right in new **Supplementary Fig. 1a**, both faster myelination of white matter with less myelin at birth (h2) and a myelin change that is independent from myelin at birth (h3) will produce a negative relation between change in T_1 and T_1 at birth. We thank the reviewer for bringing this important issue to our attention.

In a second simulation (new **Supplementary Fig. 1b**), we did the analogous analysis using R_1 rather than T_1 . In contrast to T_1 , R_1 increases linearly with myelin (new **Supplementary Fig. 1b, top left**) and a unit change in myelin leads to a constant and positive change in R_1 that is independent from the myelin fraction in a voxel (new **Supplementary Fig. 1b, top right**). As such, R_1 depends only on myelin fraction and dR_1 depends only on changes in myelin fraction. In other words, changes in R_1 as a function of R_1 at birth (new **Supplementary Fig. 1b, bottom right**) accurately reflects the hypothesized effects in myelin (new **Supplementary Fig. 1b, bottom left**). The conclusion from this simulation is hence that R_1 is an appropriate metric to distinguish developmental hypotheses, as suggested by the reviewer. Therefore, we have revised all respective figures (**Fig 2-5**) and analyses of the manuscript to use R_1 to test the developmental hypotheses. To increase awareness of these issues across the field, we also made the simulation openly available in our GitHub repository (<https://github.com/VPNL/babyWmDev>).

Supplementary Figure 1. Quantitative MRI measures of R_1 (b) but not T_1 (a) are linearly related to myelin content and changes in myelin over time. As such, R_1 is a suitable metric to distinguish between developmental hypotheses.

The new analyses using R_1 reveal that:

- (i) Mean R1 (new **Fig 2a**) as well as R1 along the length of each bundle (new **Fig 3**) increases from 0-6 months in all 24 identified white matter bundles. As R1 increases linearly with myelin fraction, this provides evidence from *in vivo* measurements that myelin content is increasing in the infant's white matter bundles, which confirms previous work.
- (ii) When evaluating mean R1 of each bundle, there is no significant relationship between the rate of development of mean bundle R1 and mean R1 measured in newborns across bundles (new **Fig 2b,c**).
- (iii) Examination of development of R1 along the length of each bundle (new **Figs 3-4**), reveals that R1 increases faster in white matter locations with lower R1 measured in newborns. These data hence provide support for the speed-up hypothesis, when evaluating white matter development along the length of the bundles. The differences between (ii) and (iii) are not surprising as white matter bundles are large structures, which have variable properties, including variable R1 at birth, along their length. Thus, mean measurements of R1 across entire bundles can obscure differential development patterns along the length of the bundle.
- (iv) Analyses of R1 development across the white matter reveal that spatial gradients are also a significant factor explaining the development of R1 during infancy.
- (v) The significance of the contribution of R1 at birth and spatial gradients are formally tested in a series of linear mixed models (LMMs) relating the rate of R1 development to R1 measured in newborns as well as spatial location in the brain. Models revealed that a significant negative relation of the rate of R1 development and R1 at birth and significant spatial gradients in the inferior-to-superior and anterior-to-posterior directions together explain myelination during early infancy. These findings are presented in new **Fig 5** and new section: **Spatial gradients and R1 at birth together explain R1 development.**

2) Another of the findings reported in the paper is that at 6 months myelination is not complete. I don't have the impression that this is a new or surprising finding as can be seen in some of the covered literature. Indeed there is literature from some of the authors of this paper showing myelination happening until the late 30's for various white matter tracts.

Thank you for pointing this out. In response, we edited the respective section of the discussion. On page 16, we now write: *"Our measurements also reveal that R1 in 6-months-olds' bundles ranges between 0.54-0.73[s⁻¹], which is lower than the average R1 measured in adults' bundles, which ranges between 0.80-1.25[s⁻¹]^{44,49}. This comparison suggests that none of the 24 bundles investigated here are fully myelinated by 6 months of age. This is not surprising, as the average R1 across the white matter develops roughly linearly during the first year of life, after which development slows down¹⁵, but continues until early adulthood^{44,50}."*

3) There is some related work that should be cited:

Soun JE, Liu MZ, Cauley KA, Grinband J. Evaluation of neonatal brain myelination using the T1- and T2-weighted MRI ratio. *Journal of Magnetic Resonance Imaging*. 2017;46(3):690–6.

Eminian S, Hajdu SD, Meuli RA, Maeder P, Hagmann P. Rapid high resolution T1 mapping as a marker of brain development: Normative ranges in key regions of interest. *PLoS One*, 2018; 13(6) Available from: <https://www.ncbi.nlm.nih.gov/pmc/articles/PMC6002025/>

Also the findings in these other papers:

Deoni SCL, Dean DC, O'Muircheartaigh J, Dirks H, Jerskey BA. Investigating white matter development in infancy and early childhood using myelin water fraction and relaxation time mapping. *NeuroImage*. 2012 Nov 15;63(3):1038–53.

Schneider J, Kober T, Graz MB, Meuli R, Hüppi PS, Hagmann P, et al. Evolution of T1 Relaxation, ADC, and Fractional Anisotropy during Early Brain Maturation: A Serial Imaging Study on Preterm Infants. *American Journal of Neuroradiology*. 2016 Jan 1;37(1):155–62.

Should be compared to the current findings to see if the question asked here was not already virtually answered in previous literature.

Thank you for suggesting these references. We have now cited them in our manuscript. It is important to note, though, that the questions addressed in our study have not been answered in prior research. First, none of the above studies examined the age range investigated in the current study. The closest is the Deoni et al. 2012 study. However, this study does not have a newborn timepoint and hence cannot test the first-in/first-out or the speed-up hypotheses, which we tested here. Second, the current study goes beyond prior work at a conceptual level, by formulating and quantitatively testing multiple developmental hypotheses. Third, none of the above studies combined quantitative MRI with dMRI tractography and thus, they are not able to assess development of specific white matter bundles, which we have done here and which is critical for testing the developmental hypotheses.

The above studies are cited as follows:

(1) Soun 2017 examined 10 neonates using the T2w/T1w ratio and concluded that T1/T2 ratio may be used as a method to contrast more myelinated vs less myelinated tissue. We now cite this study [Reference Nr 14] on page 3, where we write: *“Both of the above hypotheses build on the observation that myelin content is not homogenous in the newborn brain^{2-5,14}.”* and on page 4, where we write: *“Thus, diffusion metrics do not provide direct measures of myelination. However, quantitative MRI^{9,14,15,18,26-30} (qMRI) measurements, such as the longitudinal relaxation rate, R1 [s⁻¹], now offer metrics that are directly related to myelin content in the white matter.”*

(2) Eminian 2018 examined development in 1-20 year-olds, and reported that changes in T1 can be observed through early adulthood. We now cite this study [Reference Nr 50] on page 16, where we write: *“Our measurements also reveal that R1 in 6-months-olds’ bundles ranges between 0.54-0.73[s⁻¹], which is lower than the average R1 measured in adults’ bundles, which ranges between 0.80-1.25[s⁻¹]^{44,49}. This comparison suggests that none of the 24 bundles investigated here are fully myelinated by 6 months of age. This is not surprising, as the average R1 across the white matter develops roughly linearly during the first year of life, after which development slows down¹⁵, but continues until early adulthood^{44,50}.”*

(3) Deoni 2012 was cited in the original submission of this manuscript [Reference Nr 15] and we now refer to it in more detail, comparing their findings to our work. The study examined cross-sectional development across the lifespan from 3 months to 5 years of age. While the study could not distinguish between the start-first/ finishes-first hypothesis and the speed-up hypothesis (as there are no measures of myelin content in newborns), it did show spatial variations in white matter myelination, which inspired us to add the spatial gradient hypothesis to our manuscript in the current revision. Critically, we show that spatial gradients, together with myelination at birth, best explain myelin development during early infancy. We now cite Deoni 2012 as follows:

On page 3 we write: *“The spatial-gradient hypothesis suggests that postnatal myelination progresses in a spatially organized manner^{5,15}. Different spatial gradients of myelination have been proposed including that white matter myelination originates in neurons and follows the direction of information flow⁴ or that it occurs along a proximal to distal axis across the brain⁵.”*

On page 4 we write: *“Thus, diffusion metrics do not provide direct measures of myelination. However, quantitative MRI^{9,14,15,18,26-30} (qMRI) measurements, such as the longitudinal relaxation rate, R1 [s⁻¹], now offer metrics that are directly related to myelin content in the white matter.”*

On page 16 we write: *“Our measurements also reveal that R1 in 6-months-olds’ bundles ranges between 0.54-0.73[s⁻¹], which is lower than the average R1 measured in adults’ bundles, which ranges between 0.80-1.25[s⁻¹]^{44,49}. This comparison suggests that none of the 24 bundles investigated here are fully myelinated by 6 months of age. This is not surprising, as the average R1 across the white matter develops roughly linearly during the first year of life, after which development slows down¹⁵, but continues until early adulthood^{44,50}.”*

On page 17 we write: *“This spatial pattern differs from observations made in preterm newborns before 40 weeks of gestation, that showed fastest development in the central white matter⁴⁸. Instead, this pattern is more aligned with spatial gradients observed later in infancy and early childhood¹⁵.”*

(4) Schneider 2016 was cited in the original submission and examined white matter in preterm newborns. Critically, it is an open question if the development of white matter is the same or different in preterm compared to full term infants. In fact, the authors themselves suggest that preterm birth affects the trajectory of brain development. As such, the Schneider 2016 study evaluates an important clinical population but holds limited information for the development of typical, full-term infants, which was the focus of our work. Nonetheless, it is interesting to compare R1 values measured in preterm infants to those in full-term newborns and we hence cite the study [Reference Nr 47] accordingly in the discussion:

On pages 15-16, we write: *“Crucially, as quantitative R1 measures are comparable across MRI scanners of the same field strength^{9,15,26}, we can compare our R1 measurements in infants to those of other populations. For example, we find that R1 in white matter bundles of full-term newborns ranges between 0.42-0.55[s⁻¹], which is higher than R1 in the white matter of preterm newborns, which ranges between 0.29-0.36[s⁻¹]⁴⁸.”*

Further, on page 17 we write: *“This spatial pattern differs from observations made in preterm newborns before 40 weeks of gestation, that showed fastest development in the central white matter⁴⁸. Instead, this pattern is more aligned with spatial gradients observed later in infancy and early childhood¹⁵.”*

Minor points

4) Line 313 It is mentioned that SPGR data is acquired together with the Inversion recovery to compute the relaxation times, but the methods only mention the use of the inversion recovery sequence.

Thank you for pointing this out. The SPGRs were used only for generating the synthetic T1 images, which, in turn, were used to generate gray matter / white matter segmentations. We have corrected this in the method section, on page 21, where we now write: *“An inversion-recovery EPI (IR-EPI) sequence was used to estimate R1 relaxation time (R1) at each voxel. Spoiled-gradient echo images (SPGRs) were used together with the EPI sequence to generate whole-brain synthetic T1-weighted images.”*

5) The novelty of the methods used for defining the fiber bundles are relatively standard and should therefore be tuned down.

Thank you. Please note that tractography in infants is a nascent field associated with unique challenges because of the infant's immature white matter, unique brain morphology (it is much more condensed than the adult brain), and overall smaller brain size. While there are automated fascicle quantification tools for adults and children (e.g., Yeatman 2012, Garyfallidis 2018, Kruper 2021), there are no such automated tools for defining bundles within individual infants' brains, which is crucial for any future diagnostic method for infants. The novelty of our approach is that it (i) is a fully automated approach optimized for the infant brain that identifies 24 major white matter bundles in individual infant's native brain space, (ii) enables analyzing white matter properties and changes of these properties along the length of each bundle, and (iii) it outperforms current state-of-the-art tools developed for adults (AFQ, Yeatman 2012). We believe that this is a valuable contribution to the field, which will facilitate future research on white matter development during a critical period of human development. This view has also been emphasized by reviewer 2, who writes *“The MRI methods employed are very impressive, the resulting tractography in individual babies is excellent. This is a very difficult feat in 3/6 month olds.”*

Reviewer #2 (Remarks to the Author):

This technically impressive paper looks at the rate of myelination of white matter in the first 6 postnatal months using a longitudinal design. The authors segregate fibre bundles according to whether they are association, projection or callosal using an automated pipeline and use T1 relaxation time as a proxy of myelin content.

The MRI methods employed are very impressive, the resulting tractography in individual babies is excellent. This is a very difficult feat in 3/6 month olds. The tractometry approach in figure 4 is really interesting and tells a lot about the tracks change over time. It would be really interesting to see what the slopes are at the cross sections of overlapping tracks - e.g. the cortico spinal track has a higher rate of change in superior regions where it intersects with other white matter bundles.

Thank you for your detailed and thoughtful feedback. In regard to your specific suggestion, we agree that how developmental slopes change in regions where different bundles overlap is an interesting question. In the current revision, we included a new hypothesis, the spatial gradient hypothesis, and to test said hypothesis we took a closer look at the spatial layout of developmental slopes across nodes from different bundles (**new Fig. 5**). This figure offers some support for the notion that crossing other bundles may impact R1 slopes. For example, R1 slopes of the cortico-spinal tract (CS) are steeper where it crosses the superior longitudinal fasciculus (SLF) and arcuate fasciculus (AF). However, as we assess white matter development in 24 specific bundles rather than the entire white matter connectome, it limits our ability to identify all crossing fibers and address this question. As such, we now highlight the question of development in regions where bundles cross as an interesting direction for future work. On page 17 of the discussion, we write: *“Further, the spatial precision afforded by our methods may facilitate future work on additional spatial aspects of the development of R1 and diffusion metrics. For example, it would be interesting to formally assess if and how these measures change in spatial locations where white matter bundles cross each other.”*

1) My main criticism is with the central claim of the paper, that least myelinated white matter develops fastest in a catch up way. I have no problem as to whether it does or not, it's just that the age range sampled (0-6 months) and the size of the sample (only 6 with all three timepoints and some with only 1 timepoint from what I can read in the methods) make it impossible to say this.

Earlier work looking at relaxometry mapping with T1 would indicate effectively linear growth of R1 until ~18 months (Deoni et al 2012, Neuroimage) and I feel you need to follow to at least 12 to support the title of the paper. Moreover, from the authors plots in Figure 2, the corticospinal track still has lower T1 than all other bundles at 6 months, reinforcing that fast developing tracts haven't caught up yet.

Thank you, this comment resonated with us. In fact, it wasn't our intention to claim that all bundles have caught up by six months. Rather we only concluded that bundles that are less myelinated at birth develop faster than more myelinated bundles at birth. We initially used the term *catch-up* as it has been used previously (e.g., Dubois et al., 2016), but we are not tied to this nomenclature. As such, we have revised our manuscript and removed all references to “catch-up” from the manuscript and the title. The *catch-up hypothesis* is now referred to as the *speed-up hypothesis*, as this better captures the hypothesis we are testing and is directly supported by our data. We also extended the discussion and further highlight the Deoni study [**Reference Nr 15**] to clarify that we expect that white matter bundles continue to myelinate after 6 months of age. On page 17, we write: *“Our measurements also reveal that R1 in 6-months-olds' bundles ranges between 0.54-0.73[s⁻¹], which is lower than the average R1 measured in adults' bundles, which ranges between 0.80-1.25[s⁻¹]^{44,49}. This comparison suggests that none of the 24 bundles investigated here are fully myelinated by 6 months of age. This is not surprising, as the average R1 across the white matter develops roughly linearly during the first year of life, after which development slows down¹⁵, but continues until early adulthood^{44,50}.”*

2) I appreciate the argument that standard deviation across T1 of the tracks is reducing as you move up in age but as this is reflected in a more flat contrast on a T1 generally. The correlation in Figure 2b is also

partly driven by projection bundles having such markedly distinct T1 at birth, in their absence I'm not sure the correlation would be there.

Thank you for your comment. In response to your concern, we have removed the analysis of the standard deviation from the manuscript. Further, in response to Reviewer 1's first comment, we have reanalyzed all data and now present R1 data rather than T1, as a change in R1 is linearly related to a change in myelin content (see simulations in response to Reviewer 1, comment 1). With this new analysis, we found that, when comparing mean R1 across bundles, there is no significant correlation between the rate of mean bundle R1 development and mean bundle R1 at birth (**Fig 2c**). However, bundles are large inhomogeneous structures and there is substantial variability in both the initial R1 and the rate of R1 development across the length of the bundles (new **Fig. 3** and **Fig. 4**), which suggests that mean measurements may not be sensitive enough to determine developmental effects. Indeed, when analyzing developmental effects along the length of the bundles, we find a significant negative relationship between the rate of R1 development and R1 at birth, supporting the speed-up hypothesis. Additionally, we find that spatial gradients are also a significant factor explaining the development of R1. These findings are summarized in new **Figs 4-5** and new section **Spatial gradients and R1 at birth together explain R1 development**.

3) The discussion mentions that the data will help interpret developmental trajectories of diffusion metrics - this data is here why not show it?

Thank you for this comment. As the goal of the study was to examine the development of white matter myelin, we initially focused on quantitative measurements that are directly related to myelin fraction in the white matter (e.g., Stuber 2014). After all, diffusion metrics of the white matter such as fractional anisotropy (FA) and mean diffusivity (MD) depend not only on myelin but also on other properties that may also develop during infancy, such as fiber diameter, packing, and orientation. Thus, making inferences from these metrics to specific biological mechanisms is complicated. Nonetheless, we agree with the reviewer that we have the diffusion metrics and providing information about their development is informative. Thus, we added new analyses of MD that mirror all the analyses performed with R1. We focus on MD, as MD is less influenced by fiber orientation compared to FA and generally decreases with increasing myelin content. These new analyses include measurements of the development of mean bundle MD (**Supplementary Fig 5**), development of MD along the length of each bundle (**Supplementary Fig 6-7**), and quantitative analyses of the effect of MD at birth and spatial gradients on the development rate of MD across the white matter (**Supplementary Fig 8**).

Results highlight both similarities and differences between MD and R1 development: (i) Consistent with the idea that white matter myelinates, MD decreases significantly from 0 to 6 months both at the level of mean per bundle (**Supplementary Fig 5**) and along the bundle. (ii) Different from R1, when evaluating mean MD per bundle, there is a significant negative correlation between mean MD at birth and rate of mean MD development (**Supplementary Fig 5b,c**). As MD depends not only on myelin fraction, this correlation may be driven by other factors beyond myelination. (iii) Consistent with R1 measurements, we find that both MD at birth and spatial gradients contribute to the development of MD during early infancy (**Supplementary Fig 8**). This differential development of MD and R1 is consistent with prior reports across the lifespan (Yeatman et al., 2014) and suggests that other changes to the white matter beyond myelination contribute to MD development in the first 6 months of life. These differences also highlight the value of quantitative measures that are directly linked to myelin content to evaluate the development of white matter myelination.

Results of MD analyses are reported in each corresponding results section, and we summarize the results in the discussion. On page 15, we write: *“Interestingly, the observed developmental pattern of MD showed both similarities and differences from developmental pattern of R1. Consistent with the notion that increases in myelin (and R1) would be associated with decreases in MD, we find that MD in the white matter decreases during infancy, as reported previously⁴⁵⁻⁴⁷. However, we also find that the*

rate and pattern of MD and R1 development across the white matter are not identical. As MD is impacted by structural components of the white matter beyond myelin (e.g., fiber diameter and packing^{18,23-25}) these differences (i) highlight the importance of using measures such as R1 which are linearly related to myelin^{26,29-31} to assess myelin development specifically, and (ii) suggest that additional properties of white matter bundles beyond myelin are also developing during early infancy. Future histological measurements in postmortem pediatric samples may elucidate these mechanisms.”

4) They also mention that rate of change may be functionally relevant - there are a few papers from the Brown group that look specifically at rate of change and cognitive ability. It's a different measure of myelin but is strongly related to T1. According to those papers, rate of myelination is functionally relevant.

Thank you for highlighting this additional literature. We have added two references to the discussion on the functional relevancy of myelination during development (Deoni et al., 2016 and Chavelier et al., 2015). In the discussion we now write on page 16: *“Second, as previous data has shown a link between cognitive development, processing speed and myelin development during infancy and early childhood^{51,52}, we further hypothesize that the observed negative relationship between myelination at birth and the rate of myelin development is functionally relevant. For example, one consequence of this developmental trajectory is that it generates a more uniform distribution of myelin across the white matter, which may allow more coordinated and efficient communication across the brain.”*

REVIEWER COMMENTS

Reviewer #1 (Remarks to the Author):

I would like to thank the authors for so thoroughly and enthusiastically having addressed the comments given on the first round / previous submission.

I still have the impression that the quality of the data processing performed is state-of-the-art. Furthermore, the new analysis of MD and R1 truly gives this manuscript a further added value.

The extremely short time span studied comes with some opportunities (like ability of trying to model the change of R1 as a linear function), but makes most conclusions on the implications of the "new" findings extremely exploratory. In this context I would like to acknowledge the authors from reducing some of their claims.

The new work shown in supplemental figures 5-8 seems to be as relevant to the understanding of tissue maturation as the measurement of R1. The MD actually fits better with the model proposed by the authors in the first submission. This makes it less clear why one is explored in the main paper and the other simply left in the supplemental material. The paper would benefit to address tissue maturation, with myelination being only one of the aspects associated with it. While I agree that R1 is very sensitive to myelin, in the absence of large quantities of the former it will be modulated by whatever is in the tissue lattice.

The authors have addressed many of the concerns and I would like to suggest some relatively minor changes and analysis.

Line 100: When describing the spatial gradient hypothesis: "Finally, the spatial gradient hypothesis predicts spatial differences in the development of R1 across the white matter, that cannot be explained by differences in R1 values in newborns." This definition falls short of what is later modelled and analyzed. The gradient hypothesis, as I understood it, suggests that the evolution is partially explained by the state of "neighboring" regions (in the fiber space or brain space) and it should tend to generate less varying properties over the brain (as if myelination was flowing through the brain stem and slowly diffusing over the whole brain).

Regarding the use of the linear model over this period of six months. Looking at previous population studies: Kühne et al, (2021) Assessment of myelination in infants and young children by T1 relaxation time measurements using the magnetization-prepared 2 rapid acquisition gradient echoes sequence, *Pediatric Radiology*, 51, pages 2058–2068 (2021), the use of a linear evolution would be justified as from birth to 10 years old R1 seems to follow an exponential like behaviour $a + b(1 - \exp(-\text{Rate} \times \text{age}))$ with the rate having a time constants $\gg 6$ months. On the other hand, looking at figure 1, it would be interesting to evaluate if higher order changes are observed (visually there seems to be a tendency for the changes from 3-6 months to be larger than 0-3 months), which could suggest an acceleration of tissue maturation with age.

Line 165 "To relate our findings to previous work that evaluated diffusion metrics".... A reference would be useful to describe what works it is referring to.

Line 177: "The differential development of MD and R1 is consistent with prior reports across the lifespan and suggests that other changes to the white matter beyond myelination contribute to MD development in the first 6 months of life." The same could be said regarding R1. Particularly in early infancy, when there is very little myelin (far from the adult situation where it represents 80% of dry weight) other contrast mechanisms could also contribute significantly to R1 (such as water mobility, tissue density).

Figure 2b in the legend the VOF for example seems to be missing this might be good to include and even highlight as it is one of the fibers discussed specifically in figure 3.

Figure 3 the plot of IFOF shows an interesting behaviour that is not discussed and that seems to go against the gradient hypothesis. There is an inversion of the R1 profile from 0 to 6 months. Could it be that those middle regions are partially contaminated by crossing fibers. If there are

regions in the plots where there is a significant amount of crossing fibers, this could be coded differently as in these regions the relaxation values are coding two different populations of fibres.

Figures 3 and 4 both the forceps major and forceps minor have a rather high values in the CC and very low change rate. Could this be due to the low resolution of the T1 maps (2mm isotropic), which could add some partial volume artifacts in these region? Figure 1 could show representative R1 maps at the 3 time points for one subject. or Figure 3 could use R1 maps as background image for the fibers.

The data shown in Figures 3 and 4 is relatively redundant. Figure 3 could be sent to supp material and Figure 7 into the main manuscript.

Line 198 "Results reveals two main findings: (i) LMM slopes are positive throughout, indicating that R1 increases from birth to 6 months of age. (ii) In all bundles, there is a nonuniform rate of R1 development along the length of the bundle." The first should not be described as a finding as even the authors later acknowledged that it was expected and previously reported. Thus it could read "In all bundles, T1 increases in a nonuniform rate long the length of the bundle."

Line 217 "Different than R1, (i) MD decreases with age (Supplementary Fig. 6), and (ii) the rate of MD development along the bundles shows a spatially distinct pattern compared to R1 (Supplementary Figure 7)." A simple metric to compare MD vs R1 and dMD/dt vs $dR1/dt$ would be to in each plot refer the correlation between the two metrics. $Abs(Corr) < 0.5$ would support the different pattern comment.

Line 300 "we find that R1 in white matter bundles of full-term newborns ranges between 0.42-0.55[s⁻¹], which is higher than R1 in the white matter of preterm newborns, which ranges between 0.29-0.36[s⁻¹]⁴⁸. This observation suggests that at birth there is some level of myelin in all 24 bundles investigated here, contrasting with classic histological studies which reported myelin only in a handful of white matter bundles in newborns (e.g., the cortical-spinal tract)²⁻⁵." Two points regarding this statement: quantitative values are nevertheless very measuring method dependent (The values measured by Schneider et al using the 3D MP2RAGE might be very different from what would be measured using the IR 2D EPI sequence used in this study); in the absence of myelin other aspects can contribute to the R1 change.

Reviewer #2 (Remarks to the Author):

I thank the authors for their careful responses to our concerns. For me, one aspect still stands out. When the authors are referring to increased rates of change along bundles as a function of initial R1, does this not just reflect bundle terminations at grey matter which myelinates later irrespective of the bundle?

Otherwise only two small comments:

I appreciate there is limited space but there is data on longitudinal associations between myelin and cortex: Deoni, S.C., Dean III, D.C., Remer, J., Dirks, H. and O'Muircheartaigh, J., 2015. Cortical maturation and myelination in healthy toddlers and young children. *Neuroimage*, 115, pp.147-161.

Focused automatic tractography is a new field but there are other papers focused specifically on automated tractography in infants - this is referenced with respect to preprocessing but not to tractography (fig 12 in the paper): Bastiani, M., Andersson, J.L., Cordero-Grande, L., Murgasova, M., Hutter, J., Price, A.N., Makropoulos, A., Fitzgibbon, S.P., Hughes, E., Rueckert, D. and Victor, S., 2019. Automated processing pipeline for neonatal diffusion MRI in the developing Human Connectome Project. *NeuroImage*, 185, pp.750-763.

We thank both reviewers for their positive assessment of our work. Both reviewers commented that the previous revision has addressed their major concerns. We have now addressed all remaining minor comments. Point-by-point answers to the reviewers' comments are written in green below each comment.

REVIEWER COMMENTS

Reviewer #1 (Remarks to the Author):

I would like to thank the authors for so thoroughly and enthusiastically having addressed the comments given on the first round / previous submission.

I still have the impression that the quality of the data processing performed is state-of-the-art. Furthermore, the new analysis of MD and R1 truly gives this manuscript a further added value.

The extremely short time span studied comes with some opportunities (like ability of trying to model the change of R1 as a linear function), but makes most conclusions on the implications of the "new" findings extremely exploratory. In this context I would like to acknowledge the authors from reducing some of their claims.

The new work shown in supplemental figures 5-8 seems to be as relevant to the understanding of tissue maturation as the measurement of R1. The MD actually fits better with the model proposed by the authors in the first submission. This makes it less clear why one is explored in the main paper and the other simply left in the supplemental material. The paper would benefit to address tissue maturation, with myelination being only one of the aspects associated with it. While I agree that R1 is very sensitive to myelin, in the absence of large quantities of the former it will be modulated by whatever is in the tissue lattice.

The authors have addressed many of the concerns and I would like to suggest some relatively minor changes and analysis.

Thank you for your thoughtful review and helpful comments as well as your enthusiasm. We address all the remaining minor concerns below.

Line 100: When describing the spatial gradient hypothesis: "Finally, the spatial gradient hypothesis predicts spatial differences in the development of R1 across the white matter, that cannot be explained by differences in R1 values in newborns." This definition falls short of what is later modelled and analyzed. The gradient hypothesis, as I understood it, suggests t the evolution is partially explained by the state of "neighboring" regions (in the fiber space or brain space) and it should tend to generate less varying properties over the brain (as if myelination was flowing through the brain stem and slowly diffusing over the whole brain).

Thank you for your comment. To address this specific concern, we have revised this hypothesis on line 100 and now write: *“Finally, the spatial gradient hypothesis predicts spatially continuous differences in the development of R1 across the white matter, that cannot be explained by differences in R1 values in newborns.”*

Regarding the use of the linear model over this period of six months. Looking at previous population studies: Kühne et al, (2021) Assessment of myelination in infants and young children by T1 relaxation time measurements using the magnetization-prepared 2 rapid acquisition gradient echoes sequence, *Pediatric Radiology*, 51, pages 2058–2068 (2021), the use of a linear evolution would be justified as from birth to 10 years old R1 seems to follow an exponential like behaviour $a + b(1 - \exp(-\text{Rate} \times \text{age}))$ with the rate having a time constants $\gg 6$ months. On the other hand, looking at figure 1, it would be interesting to evaluate if higher order changes are observed (visually there seems to be a tendency for the changes from 3-6 months to be larger than 0-3 months), which could suggest an acceleration of tissue maturation with age.

Thank you. As you have already pointed out, based on the literature, e.g., Kühne et al. (2021) and also Deoni et al. (2012), the white matter is thought to develop roughly linearly during the first year of life, after which development slows down. As such, this literature predicts that, even if development were nonlinear during the 0-6 months period, the rate of development would be smaller from 3-6 months than from 0-3 months, which contrasts with your observation. Nonetheless, to address this concern more explicitly, we compared our linear models (gray line in the figure below) with a nonlinear model (red curve in the figure below). To account for your prediction for larger changes between 3-6 months than 0-3 months, we fit a second order polynomial to the R1 data. Model comparison using the Akaike Information Criterion did not provide support for the second order polynomial model being a better model than the linear model (all AIC differences < 2), suggesting that the simpler, linear model should be chosen. In addition, we have now cited the Kühne et al. (2021) paper. In the discussion on page 17, we write: *“This comparison suggests that none of the 24 bundles investigated here are fully myelinated by 6 months of age. This is not surprising, as the average R1 across the white matter develops roughly linearly during the first year of life, after which its development slows down^{15,50}, but continues until early adulthood^{44,51}.”*

Line 165 “To relate our findings to previous work that evaluated diffusion metrics” A reference would be useful to describe what works it is referring to.

Thank you for highlighting this omission. We have added references to this sentence on line 170, as follows: “To relate our findings to previous work that evaluated diffusion metrics^{17–22}, we also measured the development of mean diffusivity (MD) across bundles.”

Line 177: “The differential development of MD and R1 is consistent with prior reports across the lifespan and suggests that other changes to the white matter beyond myelination contribute to MD development in the first 6 months of life.” The same could be said regarding R1. Particularly in early infancy, when there is very little myelin (far from the adult situation where it represent 80% of dry weight) other contrast mechanisms could also contribute significantly to R1 (such as water mobility, tissue density).

Thank you for highlighting this point. This comment resonated with us, and we believe that it warrants a short discussion. As such, in the discussion, on page 17, we now write “This observation suggests that at birth there is some level of myelin in all 24 bundles investigated here, contrasting with classic histological studies which reported myelin only in a handful of white matter bundles in newborns (e.g., the cortical-spinal tract)^{2–5}. These contrasting results may be due to two reasons: On the one hand, as classic dissection studies used qualitative visual inspection of myelin stains in postmortem tissue, quantitative R1 measurements may simply be more sensitive to minimal amounts of myelin. On the other hand, more work is needed to elucidate what impacts R1 in the white matter bundles of the infant brain. While in the adult brain 90% of the variance in R1 in white matter bundles is related to myelin^{29,31}, in the sparsely myelinated infant brain, additional factors such as tissue density (e.g. proliferation of glia cells), water mobility, or changes in iron may contribute more strongly to R1.”

Figure 2b in the legend the VOF for example seems to be missing this might be good to include and even highlight as it is one of the fibers discussed specifically in figure 3.

Thank you for your comment. The reason that the VOF is not included in this figure is because the methodological approach for identifying it differs from all other bundles, both in AFQ and babyAFQ. As

the VOF is a relatively short bundle, it is difficult to identify using a pair of waypoint ROIs placed in the white matter volume, as the potential waypoints would be proximal. For this reason, the VOF is identified using a cortical surface-based ROI of ventral temporal cortex (VTC) in combination with the restriction that the streamlines have to progress vertically. In contrast to the volumetric way points for the other bundles, which are defined either in adult (AFQ) or infant (babyAFQ) templates, the cortical surface definition of the VTC is defined in each individual subject's cortical space in both AFQ and baby AFQ. Thus, as the methods for defining the VOF are different than the ones used to define other bundles, we did not include the VOF in Fig 1b.

We note that cortical segmentations are particularly challenging in young infants. Thus, to enable identifying the VOF without necessitating a cortical surface reconstruction, in babyAFQ we added another, alternative approach to defining the VOF that does not require a cortical surface reconstruction. This alternative approach uses a volumetric ROI of VTC to identify the VOF when cortical surface segmentations are not available. However, this approach is not available in AFQ and therefore we could not make this comparison and add it to Figure 1b. Nonetheless, in **Supplementary Fig 4**, we compared with babyAFQ the surface vs volumetric based approaches of defining the VOF in infants and found a high overlap in the VOF identified using the new volumetric approach compared to the classic, surface-based approach.

We now clarify this in the results section, where we write, on page 5: *"We optimized babyAFQ for infants by: [...] (iv) offering a volumetric approach for the identification of the VOF (**Supplementary Fig 4**), as the VOF is often identified using cortical surface ROIs and cortical surface reconstructions can be difficult to generate for infant brains."*

Further, also in the results section, on page 7, we now write: *The VOF and MLF were not included in this comparison to manual bundles; this is because the MLF is not identified by AFQ and the VOF is identified using a different methodological approach in AFQ (for details see **Supplementary Fig 4**).*"

Figure 3 the plot of IFOF shows an interesting behaviour that is not discussed and that seems to go against the gradient hypothesis. There is an inversion of the R1 profile from 0 to 6 months. Could it be that those middle regions are partially contaminated by crossing fibers. If there are regions in the plots where there is a significant amount of crossing fibers, this could be coded differently as in these regions the relaxation values are coding two different populations of fibres.

Thank you for pointing this out. We agree that crossing fiber regions are interesting, and it is certainly possible that the IFOF is crossing another bundle in those middle regions. However, please note that almost all bundles will be crossed by other bundles at some point along their trajectory. Disentangling the effects of crossing fiber regions is beyond the scope of the current study. To address this comment, we hence highlight this issue as a fruitful avenue for future studies. In the discussion, on page 19, we write: *"Further, the spatial precision afforded by our methods may facilitate future work on spatial dependency of both quantitative and diffusion metrics. For example, it would be interesting to formally assess if and how these measures change in spatial locations where multiple bundles cross each other."*

Figures 3 and 4 both the forceps major and forceps minor have a rather high values in the CC and very low change rate. Could this be due to the low resolution of the T1 maps (2mm isotropic), which could

add some partial volume artifacts in these region? Figure 1 could show representative R1 maps at the 3 time points for one subject. or Figure 3 could use R1 maps as background image for the fibers.

Thank you for your comment. We were concerned about partial volume artifacts from the ventricles when we set up our pipeline, as well. To address this concern, we used the brain segmentation to mask the ventricles out of the R1 and MD maps. We now added this information to the method section, on page 22, where we write: *“We also identified the ventricles in each infant and removed them from the R1 and MD maps, to limit the impact of partial volume artifacts between ventricles and white matter in neighboring bundles.”*

The data shown in Figures 3 and 4 is relatively redundant. Figure 3 could be sent to supp material and Figure 7 into the main manuscript.

Thank you. We believe it is necessary to show the raw R1 data (Fig 3) in order for the readers to understand the model fits (Fig 4). Nonetheless, we agree that it would be beneficial to move some of the MD results into the main manuscript. As such, we have integrated supplementary figures into the main figures. Specifically, we now show MD data in Figure 2 (new panels d,e) and in Figure 5 (new panels d,e).

Line 198 “Results reveals two main findings: (i) LMM slopes are positive throughout, indicating that R1 increases from birth to 6 months of age. (ii) In all bundles, there is a nonuniform rate of R1 development along the length of the bundle.” The first should not be described as a finding as even the authors later acknowledged that it was expected and previously reported. Thus it could read “In all bundles, T1 increases in a nonuniform rate long the length of the bundle.”

Thank you. We have revised these lines of the manuscript as recommended. On page 11, and now write: “Results reveal that in all bundles, there is a nonuniform rate of R1 increase along the length of the bundle.”

Line 217 “Different than R1, (i) MD decreases with age (Supplementary Fig. 6), and (ii) the rate of MD development along the bundles shows a spatially distinct pattern compared to R1 (Supplementary Figure 7).” A simple metric to compare MD vs R1 and dMD/dt vs dR1/dt would be to in each plot refer the correlation between the two metrics. $Abs(Corr) < 0.5$ would support the different pattern comment.

Thank you. In the previous version of the manuscript, we had presented such correlations between R1 and MD for the means of all bundles. That is, in the results section, on page 9, we write: *“Like R1, mean MD in newborns and the rate of mean MD development varied across bundles (Fig 2d,e). Interestingly, while mean MD and R1 in newborns are correlated ($R^2=0.76$, $p<0.0001$), the rates of MD and R1 development during early infancy are not correlated ($R^2=0.08$, $p=0.17$) across bundles. That is, the longitudinal developmental patterns observed using MD are different from those observed with R1.”* In addition, we now measured the correlation between R1 and MD along each bundle sampling every 10th node within each bundle. Every 10th node was chosen to ensure that nodes are not sampled from the same voxels. Results of this analysis are consistent with the analysis of the bundle means and show that R1 and MD measured in newborns are more strongly correlated to each other (all $R^2 > 0.27$ [range: 0.27-

0.96], all $p < 0.12$ [range: 0.0000005-0.12]) than the R1 and MD slopes (all $R^2 > 0.005$ [range: 0.005-0.83], $p < 0.84$ [range: 0.0002-0.84]).

Line 300 “we find that R1 in white matter bundles of full-term newborns ranges between 0.42-0.55[s-1], which is higher than R1 in the white matter of preterm newborns, which ranges between 0.29-0.36[s-1]48. This observation suggests that at birth there is some level of myelin in all 24 bundles investigated here, contrasting with classic histological studies which reported myelin only in a handful of white matter bundles in newborns (e.g., the cortical-spinal tract)2–5.” Two points regarding this statement: quantitative values are nevertheless very measuring method dependent (The values measured by Schneider et al using the 3D MP2RAGE might be very different from what would be measured using the IR 2D EPI sequence used in this study); in the absence of myelin other aspects can contribute to the R1 change.

Thank you for pointing this out. We have revised this paragraph in the discussion. On page 17, we now write: *“This observation suggests that at birth there is some level of myelin in all 24 bundles investigated here, contrasting with classic histological studies which reported myelin only in a handful of white matter bundles in newborns (e.g., the cortical-spinal tract)2–5. These contrasting results may be due to two reasons: On the one hand, as classic dissection studies used qualitative visual inspection of myelin stains in postmortem tissue, quantitative R1 measurements may simply be more sensitive to minimal amounts of myelin. On the other hand, more work is needed to elucidate what impacts R1 in the white matter bundles of the infant brain. While in the adult brain 90% of the variance in R1 in white matter bundles is related to myelin^{29,31}, in the sparsely myelinated infant brain, additional factors such as tissue density (e.g. proliferation of glia cells), water mobility, or changes in iron may contribute more strongly to R1.”*

Reviewer #2 (Remarks to the Author):

I thank the authors for their careful responses to our concerns. For me, one aspect still stands out. When the authors are referring to increased rates of change along bundles as a function of initial R1, does this not just reflect bundle terminations at grey matter which myelinates later irrespective of the bundle?

Thank you for your feedback and thoughtful review. We would expect any impact of changes in the cortical grey matter to be restricted to the very first and very last 1-2 nodes of each bundle, as only these nodes reach cortex. As such, this concern can be addressed in a control analysis, where the first and last few nodes (we choose 10 nodes at the ends of each bundle to be conservative) are excluded. We implemented this control analysis for the final combined model (Figure 5) that simultaneously tests for an impact of R1 at birth and spatial gradients and found similar results as in our main model. We describe the results of this new control analysis on page 15, where we write: “Significant effects of R1 measured in newborns ($\beta = -0.0012$; $p = 0.006$) and spatial location (z axis: $\beta = 1.21 \times 10^{-4}$, $p < 0.0001$, y axis: $\beta = -1.19 \times 10^{-4}$, $p < 0.0001$, y*z axis: $\beta = 1.79 \times 10^{-4}$, $p < 0.0001$) were also observed when the first and last 10 nodes were excluded from the model, which showed that the observed effects are not predominantly driven by nodes in proximity of the cortical gray matter.” These additional results suggest that our findings are specific to the white matter and not driven by changes in the cortical grey matter.

Otherwise only two small comments:

I appreciate there is limited space but there is data on longitudinal associations between myelin and cortex: Deoni, S.C., Dean III, D.C., Remer, J., Dirks, H. and O’Muircheartaigh, J., 2015. Cortical maturation and myelination in healthy toddlers and young children. *Neuroimage*, 115, pp.147-161.

Thank you for pointing out this relevant literature. We now refer to this study [reference 55] in the discussion, on page 18, where we write: *“We hypothesize that the consequence of these spatial gradients may be to allow white matter that supports crucial functions such as vision (occipital lobe) and motor control (parietal lobe) to develop faster during infancy. Another interesting avenue for future studies could hence be to examine the relationship between R1 development in the white matter and R1 development in cortex^{54,55}.”*

Focused automatic tractography is a new field but there are other papers focused specifically on automated tractography in infants - this is referenced with respect to preprocessing but not to tractography (fig 12 in the paper): Bastiani, M., Andersson, J.L., Cordero-Grande, L., Murgasova, M., Hutter, J., Price, A.N., Makropoulos, A., Fitzgibbon, S.P., Hughes, E., Rueckert, D. and Victor, S., 2019. Automated processing pipeline for neonatal diffusion MRI in the developing Human Connectome Project. *NeuroImage*, 185, pp.750-763.

Thank you for pointing this out. We have edited the section of the manuscript where we had credited this work previously [reference 70], and now cite it also in the context of tractography. On page 24 we now write: *“DMRI preprocessing and tractography was performed in accordance with recent work from the developing human connectome project^{69,70}, using a combination of tools from MRtrix3^{70,71} (github.com/MRtrix3/mrtrix3) and mrDiffusion (<http://github.com/vistalab/vistasoft>).”*

REVIEWERS' COMMENTS

Reviewer #1 (Remarks to the Author):

While the authors now refer that CSF region was masked, it is not really clear how this was performed. To avoid partial volume contamination artifacts it would be important to consider a certain degree of dilation of the CSF mask.

Reviewer #2 (Remarks to the Author):

I have no further comments. I thank the authors for their comprehensive responses to the comments.

REVIEWERS' COMMENTS

Reviewer #1 (Remarks to the Author):

While the authors now refer that CSF region was masked, it is not really clear how this was performed. To avoid partial volume contamination artifacts it would be important to consider a certain degree of dilation of the CSF mask.

Thank you again for your constructive and valuable input. To address this minor concern, we now elaborate on how the ventricles were identified and masked. On page 19 of the manuscript, we now write: "We also identified the ventricles in each infant using the iBEAT ventricle labels. We visually inspected these labels in each infant and time point and manually edited them where necessary, to ensure that all voxels that contained cerebral spinal fluid (CSF) were included in the label. We then used this label as a mask, thus removing the ventricles from the R1 and MD maps, to limit the impact of partial volume artifacts between CSF and white matter in neighboring bundles."

Reviewer #2 (Remarks to the Author):

I have no further comments. I thank the authors for their comprehensive responses to the comments.

We would again like to thank the reviewer for their helpful feedback that improved our manuscript.